



# Has dyke development in the Vietnamese Mekong Delta shifted flood hazard downstream?

Nguyen Van Khanh Triet[1,4], Nguyen Viet Dung[1,4], Hideto Fujii[2], Matti Kummu[3], Bruno Merz[1], Heiko Apel[1]

[1]GFZ German Research Centre for Geosciences, Section 5.4 Hydrology, 14473, Potsdam, Germany
[2]Faculty of Agriculture, Yamagata University, 1-23 Wakaba-machi, Tsuruoka, Yamagata 997-8555 Japan
[3]WDRG Water & Development Research Group, Aalto University, Finland
[4]SIWRR Southern Institute of Water Resources Research, Ho Chi Minh City, Vietnam

*Correspondence to*: N. V. K. Triet (triet@gfz-potsdam.de)

**Abstract**

In the Vietnamese part of the Mekong Delta (VMD) the areas with three rice crops per year have been expanded rapidly during the last 15 years. Paddy-rice cultivation during the flood season has been made possible by implementing high-dyke flood defenses and flood control structures. However, there are widespread claims that the high-dyke system has increased water levels in downstream areas. Our study aims at resolving this issue by attributing observed changes in flood characteristics to high-dyke construction and other possible causes. Maximum water levels and duration above the flood alarm level are analysed for gradual trends and step changes at different discharge gauges. Strong and robust increasing trends of peak water levels and duration downstream of the high-dyke areas are found with a step change in 2000/2001, i.e. immediately after the disastrous flood which initiated the high-dyke development. These changes are in contrast to the negative trends detected at stations upstream of the high-dyke areas. This spatially different behaviour of changes in flood characteristics seems to support the public claims. To separate the impact of the high-dyke development from the impact of the other drivers, i.e. changes in the flood hydrograph entering the Mekong Delta, and changes in the tidal dynamics, hydraulic model simulations of the two recent large flood events in 2000 and 2011 are performed. The hydraulic model is run for a set of scenarios whereas the different drivers are interchanged. The simulations reveal that for the central VMD an increase of 9–13 cm in flood peak and 15 days in duration can be attributed to high-dyke development. However, for this area the tidal dynamics have an even larger effect in the range of 19–32 cm. However, the relative contributions of the three drivers of change vary in space across the delta. In summary, our study confirms the claims that the high-dyke development has raised the flood hazard downstream. However, it is not the only and not the most important driver of the observed changes. It has to be noted that changes in tidal levels caused by sea level rise in combination with the widely observed land subsidence and the temporal coincidence of high water levels and spring tides have even larger impacts. It is recommended to develop flood risk management strategies using the high-dyke areas as retention zones to mitigate the flood hazard downstream.

## 1 Introduction

The Vietnamese Mekong Delta (VMD), the so called rice bowl of Viet Nam, encompasses an area of 4.0 million hectare, of which over 2.6 million hectare are used for agriculture. It accounts for more than 52 % of the national food production and more than 85 % of the annual rice export (GSO, 2015). Being a low-lying coastal region, the VMD is susceptible to both riverine and tidal floods, threatening agricultural production and the safety of people (Schumann et al., 2008;Kuenzer et al., 2013), (Le et al., 2007;Dung et al., 2009;Dung et al., 2011;Van et al., 2012). Climate change and sea level rise are expected to increase the risk, not only due to flooding but also due to droughts and salinity intrusion (Wassmann et al. (2004);Khang et al. (2008);Toan (2014), Hak et al. (2016)). Further, the region faces severe sediment starvation as consequence of massive hydropower development (Fu and He, 2007;Kummu and Varis, 2007;Fu et al., 2008;Kummu et al., 2010;Xue et al.,





2011;Gupta et al., 2012;Manh et al., 2015). In a recent study, Manh et al. (2015) estimated the delta sedimentation and the amount of sediment reaching the South China Sea to be diminished by 40–95 % considering hydropower development scenarios, climate change, sea level rise and the deltaic land subsidence. Land subsidence becomes a notable problem in many cities in the delta, with an estimated rate of 1–4 cm year$^{-1}$ (Erban et al., 2014;Anthony et al., 2015).

The population in the VMD has extensive experience in living with floods. The first major human interventions were the construction of the Thoai Ha and Vinh Te canals in the 1820s, followed by several water works before and during the colonial period as well as after the reunification in 1975. However, large-scale flood control systems had not been implemented until the late 1990s (Käkönen, 2008). Starting from 1996, the Government of Viet Nam constructed a series of waterways to divert part of the overland flood flow from Cambodia towards the Gulf of Thailand, followed by the development of comprehensive

dyke systems and hydraulic structures according to the Five-Year Development Plan 1996–2000 for the region (The Government of Viet Nam, 1996). The majority of dykes constructed within this period were so called low-dyke. Low-dyke provide protection against the early flood peak arriving around mid-July to mid-August, ensuring the farmers to grow two rice crops per year by keeping flood water in the paddy fields after the summer crop. The so called high-dyke were mainly built after the disastrous flood in 2000. They were designed according to farmers' needs and the demand of the provinces to protect

the floodplains against a flood as high as in 2000. This protection standard along with the full control of the flow into the floodplains allows nowadays the cultivation of three crops per year in the provinces An Giang and Dong Thap in the upper part of the delta.

The implementation of high-dyke was claimed to benefit the population by providing safety and additional income for the farmers. However, Howie (2005);Käkönen (2008) questioned this claim, because impeding floodplain inundation reduces the

input of sediments and thus of natural fertilizers to the paddy fields. This leads to reduced crop yields (Manh et al., 2014), and consequently intensified use of agrochemicals and higher costs. Howie (2005) and Käkönen (2008) also pointed to other social and environmental adverse consequences, e.g. water pollution, stress and exhaustion because of missing "resting time" for the farmers, and higher damages in case of an extreme event causing dyke breaches and flooding of the third summer crops. One important aspect in this discussion is the question to which extent upstream flood control has increased flood risk downstream.

Recently, Fujihara et al. (2015);Dang et al. (2016) attempted to quantify the impacts of the high-dyke development on downstream water levels. Both studies derived high-dyke areas from Moderate Resolution Imaging Spectroradiometer (MODIS) satellite images, and analysed historical records to detect trends in water levels and to attribute these trends to the dyke development. Fujihara et al. (2015) investigated the period 1987–2006 at 24 stations in the VMD, while flooded areas were identified for 2000–2007. Dang et al. (2016) reduced the number of stations to ten, but included four stations in the

Cambodian flood plains (CFP) and expanded the hydrological and satellite data to 2013/2014. Both studies concluded that high-dyke development in the northern part of the VMD and the increasing water levels downstream are linked. Dang et al. (2016) also stated that the hydraulic alterations in the VMD had the largest influence on water level changes, compared to dam construction in the Mekong basin, sea level rise and deltaic land subsidence. However, both studies were not able to quantify the contribution of the hydraulic alterations to the observed changes. Moreover, Dang et al. (2016) noted that the findings were

prone to considerable uncertainties, notably the uncertainty of the water level data. Possible sources of these uncertainties are land subsidence, change in reference datum, and instrument or observer errors. Building on these findings, our study has two objectives: 1) to re-analyse the water level trends found in the previous studies under explicit consideration of uncertainty, and 2) to quantify the changes caused by the development of the dyke system on water levels in the VMD by a detailed hydraulic modelling approach, as recommended by Dang et al. (2016).

The first objective is achieved by comparing flood trends at the two major gauging stations downstream of the high-dyke development areas with flood trends at stations upstream of the development areas. This comparison includes an analysis of the robustness against data errors. To obtain a better understanding of the trends, changes in flood peak and flood duration are analysed, which in combination define the severity of floods in the Mekong Delta (Dung et al., 2015). To quantify the relative





contribution of high-dyke development, we update the quasi-2D flood model developed by Dung et al. (2011);Manh et al. (2014) using the latest comprehensive dyke survey data and high-resolution topographical data from the newly available 5x5 m resolution LiDAR-DEM for the VMD. Changes in delta inundation dynamics for the pre- and post-high-dyke development are investigated in different model setups, simulating the two recent, most severe flood events in 2000 and 2011.

## 2 Study area, data and methodology

### 2.1 Study area

Originating from the Tibetan Plateau, the Mekong River runs through the territory of China, Myanmar, Laos, Thailand, Cambodia and Viet Nam before it discharges into the South China Sea (termed East Sea in Vietnam). With the mainstream length of almost 4,100 km, drainage area of about 795,000 km$^2$ and annual mean discharge of 475 km$^3$, the Mekong is ranked 10[th] in the list of the world largest rivers. The basin is subjected to a monsoonal regime, in which the wet season lasts from June to November accounting for 80–90% of the total annual flow. During this period a huge amount of water is routed to the lowland areas, causing extensive inundation in the MD. The MD encompasses the Cambodian floodplains (CFP) downstream of Kratie (Fig. 1) and the VMD, which differs considerably from the CFP due its enormous amount of man-made hydraulic structures like channels, dykes, sluice gates and pumps. Although large floods, as for example in 2000 or 2011, result in excessive economic and social damages (MRC, 2012, 2015), flooding is the backbone of the agricultural production in the region. Moderate floods, commonly perceived as "good floods" by the local population, bring various benefits, e.g. reduction of soil acidity, removal of residual pesticides and other pollutions in paddy fields, an immense wealth of wild fish in the rivers, channels and inundated floodplains, and provision of nutrients through deposited sediments on the floodplains (Hashimoto, 2001;Sakamoto et al., 2007;Hung et al., 2012;Manh et al., 2013;Manh et al., 2014).

Flood water enters the VMD via three main routes, i.e. the mainstream flow through the Mekong River (named Tien River in Vietnam) and Bassac River (named Hau River in Vietnam), and transboundary overland flow to the Plains of Reeds (PoR) east of the Mekong River and to Long Xuyen Quadrangle (LXQ) west of the Bassac River (Fig. 1). The mainstream flow accounts for 90 % of the total flood volume entering the VMD (Tri, 2012). The flood prone areas cover a territory of approximately 2.0 million hectares in the northern part of the VMD (Fig. 1). The average inundation depth varies from 0.5 to 4.0 m and lasts for 3 to 6 months (Toan, 2014;Xo et al., 2015). The floodplains are protected by extensive dyke systems, both low-dyke and high-dyke, with a total length of over 13,000 km, of which 8,000 km are low-dyke. High-dyke areas are mainly concentrated in the provinces An Giang and Dong Thap in the upper part of the delta (green areas in Fig. 1). About 65 % of the cultivation area in An Giang and 40 % in Dong Thap are protected by high-dyke with crest levels of 4.0–6.0 m.a.s.l. (SIWRR, 2010). The area of triple cropping in 2014 was 175,000 ha in An Giang and 120,000 ha in Dong Thap, respectively (GSO, 2015).

### 2.2 Data

#### 2.2.1 Gauge data

Gauge data were used as boundaries for the hydraulic model, for model calibration and validation and for detection of flood trends. Daily and hourly water level and discharge data were collected at 30 stations. They include 13 stations along the Rivers Mekong and Bassac, of which 6 stations are located in Cambodia, along with 10 tidal gauge stations and 7 stations located in secondary rivers/canals (Fig. 1). The location and name of each station are given in Table S1 (supplement). These data were provided by the Mekong River Commission (MRC) and the Southern Regional Hydro-Meteorology Centre of Viet Nam (SRHMC). The MRC database contains daily readings from 1960s to present, with some stations having data back to the 1930s. However, the data often have gaps due to the past political conflicts in the region. SRHMC data are measured in hourly intervals and are relatively complete, but cover only the period from the late 1970s to present. To obtain a common period for





the trend analyses for the Vietnamese and Cambodian stations, time series of 38 years of daily water level/discharge data (1978–2015) at 5 key stations, i.e. Kratie, Tan Chau, Chau Doc, Can Tho and My Thuan, were chosen. These two databases use different vertical reference points, i.e. Hatien1960 in MRC data, and Hondau72 in SRHMC data, respectively. To ensure consistency, the MRC data were converted to the Hondau72 datum.

For the calibration and verification of the hydraulic model, hourly data of all VMD gauging stations (Fig. 1 and Table S1 in Supplement) and inundation maps derived from MODIS satellite images were used. Cloud cover of the MODIS images was removed by Kotera et al. (2014);Kotera et al. (2016). The calibration considered the rainy season (June to November) of the years 2000 and 2011.

### 2.2.2 Dyke and topography data

To evaluate the impact of the high-dyke development, two benchmark states were selected: the year 2000, which marked the start of the large-scale high-dyke development, and the status in 2011 with large-scale high-dyke protection in place. These two years are among the most severe floods in the MD, second only to the flood in 1978. The dyke system in 2000 was taken from Hoi (2005). Details of the dyke system of 2011 were based on a survey undertaken by the Southern Institute of Water Resources Research (SIWRR) from 2009 to 2010, i.e. it included all the dykes constructed before the 2011 flood, except those

possibly built in the dry season of 2011. The survey was conducted in four flood prone provinces in the VMD, i.e. An Giang, Dong Thap, Kien Giang and Long An, and included maps of compartments protected by dykes, and their classification as either low-dyke or high-dyke. In addition, dyke height, width at crest, slopes and sluice gate locations and further specifications were provided in the survey.

The high-resolution LiDAR-based DEM surveyed during 2008-2010 for all provinces in the VMD was obtained from the

Ministry of Natural Resources and Environment of Vietnam (MONRE). Floodplain compartments with dyke-lines and elevation, which were not detectable in the Shuttle Radar Topography Mission (SRTM) DEM used in the previous version of the hydraulic model, are represented in the high resolution DEM, as the widths of dyke crests are in the range of 3.0–5.0 m. The final dyke heights of the updated hydraulic model were determined by combining information from these two sources of topographical information. The new DEM was also applied to extract updated cross-sections for the representation of

floodplain compartments and to derive simulated inundation extent and duration maps from the hydraulic model results.

### 2.3 Trend analysis

*Long-term trend detection*

The Mann−Kendall (MK) non-parametric trend test (Mann, 1945;Kendall, 1975) was applied to detect changes in flood characteristics. The trend magnitude was estimated by the non-parametric method of Sen (1968). The annual maximum water

level (AMWL) and the flood volume and duration were selected as flood characteristics, as the flood severity is not fully represented by the maximum discharge/water level alone (Dung et al., 2015). For example, the maximum water level in 2014 at Kratie was 19 cm higher than the peak of the disastrous flood in 2000. However, in 2014 the duration of flow above which overbank flow and floodplain inundation starts and the volume of the flood hydrograph were about 55 % and 65 % of 2000, respectively. This explains the large difference in losses for these two floods.

The annual flood volume (AFV), defined as the sum of daily discharge during the flood season June–November, at Kratie was selected as second important factor describing the severity of the floods entering the MD. At all downstream gauges, the number of days with maximum water level equal or higher than a certain threshold value (DOT) was selected. This indicates the severity of floods in terms of the duration of critical water levels and was chosen, instead of the annual flood volume, because discharge data are less complete, and because river flow is influenced by the tidal magnitude at the stations in the

VMD. For the coastal stations and those directly downstream of the high-dyke areas (e.g. Can Tho, My Thuan), the tidal signal



is clearly visible even during the high-flood period (Hung et al., 2012). This means that under high-tide conditions overbank flow and inundation can occur even during comparatively low river discharge.

The critical threshold values were selected on the basis of the three flood alarm levels, as indicated in the National Standard of Vietnam. When the water stage exceeds the alarm level 3, flooding reaches an unusual and hazardous state. Although level 3 is a good indicator for damaging floods, it is not suitable as threshold to calculate flood duration, as only in a few years of the study period the water level exceeded this highest alarm level. For instance, at Tan Chau and Chau Doc, level 3 was reached in only 11 years. Using this level would not yield robust results for the trend tests. Hence, the alarm level 1 was used, i.e. 3.50 m at Tan Chau, 3.0 m at Chau Doc and 1.5 m at Can Tho and My Thuan.

*Step change detection*

Past studies agreed on the direction and magnitude of flood trends at key locations in the VMD, specifically for Can Tho and My Thuan (Le et al., 2007;Fujihara et al., 2015;Dang et al., 2016). They disagreed, however, on the timing of the changes, and consequentially, on the reasons for changes. Le et al. (2007) found rising water levels at some stations in the VMD and linked these changes to infrastructure development during the period 1996–2002. Similarly, Toan (2014) reported an unprecedented change in tidal levels at Can Tho after 2000. On the other hand, Fujihara et al. (2015) concluded that infrastructure development until 2006 had minor impacts. Therefore, to analyse if and when changes in water levels can be linked to infrastructure development, the non-parametric approach by Pettitt (1979) for detecting step changes was applied in addition to the MK test. This method has not been applied by the previous studies.

*Uncertainty analysis of the detected trend*

Gauged data are subject to errors, stemming from either human operation or instrument failures including subsidence of the gauge reference. These errors might influence the trend analysis. Therefore, the trend analysis was tested against hypothetical measurement errors in a Monte Carlo (MC) experiment by randomly disturbing the observation data with uniformly distributed errors. The error ranges were ± 5 cm, ±10 cm and ± 20 cm, from which randomly drawn errors were added to every gauged water level data point. The lowest error range is derived from the accuracy range of instruments applied to monitor water stage e.g. staff level gauges, automatic measurement using sonar or pressure probes, which is in the range of ± 2 cm for the current modern instruments in place. In order to account for lower precision of earlier instruments and the unknown error range of manual staff readings this range was extended to ± 5 cm. The higher error ranges are meant for testing the trends against hypothetical higher errors not caused by instrument errors alone. For Kratie, these error ranges in water level correspond to an alteration of 1 %, 3 % and 6 % in discharge based on a rating curve derived from contemporaneous records of water level and discharge. AMWL, DOT and AFV were calculated from the disturbed time series, and the MK trend test, Sen's slope estimation and Pettitt's test were performed again. By repeating this procedure 1000 times, the robustness of the detected trends against data errors was tested.

### 2.4 Hydrodynamic modelling

To quantify the contribution of high-dyke development, a hydrodynamic model for the simulation of the flood inundation process in the MD was used. The model includes the CFP, the Tonle Sap Lake as well as the majority of the channels and hydraulic structures in the VMD. The model was initially developed by Dung et al. (2011) and refined by Manh et al. (2014). It is a quasi-2D model, and takes the complex hydraulic system of the VMD into account. A typical flood compartment, i.e. part of the floodplain encircled by channels, is described by "virtual" channels with wide cross-sections connected to the channels by sluice gate model structures. These cross-sections were originally extracted from the SRTM DEM. The cross-section width is defined in such a way to preserve the flood compartment area. Dyke-lines of each flood compartment are described by four control structures right after the points where virtual and real channels are linked. These are





introduced in the model as broad crest weirs. The crest levels of dyke-lines are presented as sill levels of these control structures (see Fig. 2). A comprehensive description of how floodplain compartments are introduced by the "virtual" channels and wide cross-sections can be found in Dung et al. (2011). The model has been calibrated by Dung et al. (2011) and Manh et al. (2014) with recent flood events in the VMD, encompassing the high floods of 2011, the medium floods in 2008 and 2009, and the

extraordinary low flood in 2010.

The multi-objective calibration by Dung et al. (2011) showed that the model could satisfyingly simulate the inundation extent only, if the model dyke levels were lowered by 20 %. This indicated that the dyke level data were not accurate. Therefore, in this study the model was updated with the latest DEM and dyke survey data as follows: first, dyke survey maps and river networks of the flood model were loaded into ArcGIS, in order to identify which and where updates in the model setup were

necessary. These were typically single big flood compartments in the current model setting, which can only be set as either low-dyke or high-dyke. However, based on the up-to-date survey data, they consist of multiple smaller flood compartments composed of low-dyke and high-dyke. Thus, the river network model for these areas needed to be updated. Figure 3 demonstrates an example. The compartment is located at Binh Phu commune in the An Giang Province. It includes four small flood compartments with both dyke types: high-dyke (number 25 and 26) and low-dyke (number 55 and 57). In the original

model setup it was represented as a single and semi-protected flood compartment (Fig. 3B). Figure 3C shows the compartment in the new model representing high- and low-dyke areas.

In total, approximately 150 flood compartments were updated. In a second step the wide cross-sections representing floodplains, which were previously extracted from the SRTM DEM, were derived from the higher resolution LiDAR-DEM. The hydrodynamic model was then calibrated against observed water levels and discharge, and inundation extents derived

from MODIS satellite images. The Nash-Sutcliffe efficiency (NSE) and the flood area index (FAI) were used to quantify model performance. Results are presented in Fig. 4 and Table 1. The majority of stations used for model calibration have NSE values in the range of 0.85 to 0.98: for the years 2000 and 2011, 11 and 14 out of 19 locations, respectively, have NSE values greater than 0.85 with a maximum value of 0.98 (Table 1). At other locations, NSE values are lower, but still higher than 0.7. These are mainly stations with tidal influence, e.g. Can Tho, My Thuan. Generally, the model captures well the water levels at high

water stages (see Fig. 4A). Its performance deteriorates for low water stages, because of a tendency to overestimate low water levels. As the focus of this study is flooding, the model performance for simulating water levels is well acceptable. The model calibration against discharge is also good, with NSE in the range 0.68–0.96. In addition, inundation extents generated from the model results are very similar to water masks derived from remotely sensing data (see Fig. 4 C and D). The good agreement is expressed by the relatively high FAI, which improved from 0.46 of the original model to 0.64 (Table 2). One possible

explanation for the mismatch of simulated and observed inundation extend for the areas in Ca Mau peninsula (the South-western part of the VMD) is the land-use type. In this area shrimp farming or a mixture of shrimp farming and paddy rice dominates throughout the year. Thus the areas detected as wet from satellite data could very likely be the water surface of shrimp ponds. As these farming schemes are not considered in the model setup, they are simulated as dry areas. In fact, this area is not known to be flooded regularly, thus this explanation is plausible. Overall and despite this mismatch of simulated

inundation areas in Ca Mau the FAI values are comparable with other flood inundation modelling studies (e.g. Aronica et al. (2002)). Therefore, it can be concluded that the updated model is reliable and produces realistic inundation dynamics, and can thus be used to evaluate the impacts of high-dyke development on flood characteristics.

## 2.5 Quantifying the contributions of dyke development and of other drivers

An inspection of the hydrographs at the tidal stations in 2000 and 2011 revealed an increase in the tidal level. At tidal gauge

My Thanh, approximately 80 km downstream of Can Tho, an increase of 22 cm in the water level averaged across the flood season was observed. Because a tidal signal can still be found at gauge stations at the Vietnam – Cambodia border (Hung et al., 2012), changes in the lower boundary might have a notable contribution to water level changes in the VMD. Also the flood





hydrograph entering the Mekong Delta at Kratie showed some notable differences between 2000 and 2011. To separate the contributions of the three drivers, i.e. changes in the upper boundary, in the lower boundary, and in the dyke system, on the flood dynamics in the VMD, the study design shown in Table 3 was developed. For each driver, the flood dynamics, i.e. inundation extent, depth and timing, were simulated for two virtual scenarios and compared to the historical situation (called the baseline scenario), i.e. the floods in 2000 and 2011. For example, to separate the impact of the dyke development, the dyke situation in 2000 (mainly low-dyke) and in 2011 (many high-dyke areas in An Giang and Dong Thap) were interchanged, whereas the other two drivers were taken from the historical situation. In other words, we simulated how the flood in 2000 would have propagated, if the high dyke system of 2011 was already in place, and how the flood in 2011 would have propagated, if the dyke system of 2000 was still be in place. The same procedure was applied for the other two drivers.

## 3 Results and discussion

### 3.1 Long-term flood trends and uncertainty analysis

A summary of the trend analysis of the five selected stations for the period 1978–2015 is presented in Fig. 5. Figure 5A presents the results for AMWL (annual maximum water level), while Figure 5B shows the results for AFV (annual flood volume) at Kratie and DOT (flow duration over threshold) at the other locations. Our findings reveal highly significant upward trends in both flood peak and flood duration downstream of high-dyke areas, i.e. in Can Tho and My Thuan. On the contrary, negative trends are detected at upstream locations, but at lower significance levels. The slopes of the downward trends in AMWL at the upstream stations Chau Doc and Tan Chau are larger than the upward trends at Can Tho and My Thuan. For DOT the upward and downward trend slopes are in the range of 1.0–1.5 day year$^{-1}$.

The results of the uncertainty analysis presented in Table 4 indicate that the upward trends at the downstream stations Can Tho and My Thuan are robust against data errors, as similar flood trends are detected at these stations for all assumed error magnitudes with high significance (p-value and hereafter termed as p ≤ 0.001). For AMWL, the median slope and even the minimum and maximum detected slopes remain similar to those of the original data set. Interestingly, the increasing detected slopes on DOT at these two stations are even higher than those of the original data. The downward trends at the upstream stations are, however, not robust against data errors, as indicated by the range of slopes, the MK test statistic and the low to non-existing overall significance in Table 4. In the following, the results of the trend analysis are discussed in detail for the different regions of the MD from upstream to downstream.

*Trends of floods entering the Mekong Delta*

The station Kratie on the Mekong mainstream represents the characteristics of the floods entering the MD. For Kratie, positive trends are obtained for AMWL and negative trends for AFV, respectively. However, these trends are very small, not statistically significant and not robust against data errors. The positive trend of AMWL, although not statistically significant, is in line with Delgado et al. (2010). They investigated the flood variability for the Mekong and found through non-stationary extreme value statistics increasing probabilities for annual maximum discharges exceeding the 2-year flood at Kratie after 1975. Prior to 1975 the trends were, however, decreasing. Using the MK test on annual maximum discharge from 1924–2007, they reported an overall negative trend. Because discharge is calculated from water level data using a rating curve, performing the MK test on either discharge or water level should yield similar results. The difference between our study and Delgado et al. (2010) can be explained by the different time periods used. The small increasing trend detected for 1978–2015 is cancelled out in the longer time series (1924–2004) by the declining trends prior to 1975.

*Trends of flood entering the Vietnamese Mekong Delta*





The stations Tan Chau and Chau Doc, where the Mekong and Bassac Rivers enter the VMD, are located just upstream of the high-dyke areas. These stations exhibit decreasing trends in both AMWL and DOT ($p > 0.1$ to $p \leq 0.05$). AMWL at Tan Chau and Chau Doc decreases by 17 mm year$^{-1}$ ($p \leq 0.1$) and 15 mm year$^{-1}$ ($p > 0.1$), respectively. Interestingly, Fujihara et al. (2015) found increasing, non-significant trends of 40.5 – 45.6 mm year$^{-1}$ ($p > 0.1$) for the period 1987–2006 for these stations.

This mismatch between our study and Fujihara et al. (2015) can be explained by the different time periods, as trend tests can be very sensitive to the selection of the time period (e.g. Hall et al., 2014), particularly at these low significance levels. This is also expressed by our test for robustness, where the assumed data errors can change the significance of the trends even for an assumed error of ± 5 cm only (Table 4). A pronounced downward trend in DOT is obtained at Tan Chau, with an estimated slope of -1.0 day year$^{-1}$ ($p \leq 0.05$). This trend is rather robust, even when data errors of ±20 cm are added. Over 96 percent of

the 1000 disturbed time series indicate a negative slope at $p \leq 0.05$, and the coefficient of variation is less than 5%. A similar trend is obtained at Chau Doc, although smaller and less robust against errors (slope: -1.0 day year$^{-1}$, $p > 0.1$).

In summary, there are weak negative trends in terms of flood peak and duration at the entrance of the VMD during the study period 1978–2015, of which the trends in DOT are more pronounced and robust. This result is comparable with the findings in (Dang et al., 2016), who showed decreasing but non-significant trends of annual mean water levels for Tan Chau and Chau

Doc.

*Trends downstream of high-dyke development areas*

Contrary to the three upstream locations, the two stations downstream of high-dyke areas, i.e. Can Tho and My Thuan, show significant upward trends in both AMWL and DOT. AMWL increases by 13.1 mm year$^{-1}$ and 9.2 mm year$^{-1}$ ($p \leq 0.001$) at Can Tho and My Thuan, respectively. This is similar to the findings of Fujihara et al. (2015). Dang et al. (2016) analysed annual

mean water levels and detected a significant upward trend at Can Tho, but no trend in My Thuan (Fig. 6). This, at first sight, contradictory result can be explained by the different relation between annual maximum and mean water levels at the two stations. There is a strong correlation between maximum and mean water levels for Can Tho ($r = 0.84$), but only a weak correlation is seen for My Thuan (Figure 6). The MK test for DOT indicates upward trends at $p \leq 0.001$, whereas the slope varies from 0.9 day year$^{-1}$ at My Thuan (Mekong River) to 1.6 day year$^{-1}$ at Can Tho (Bassac River). Both trends in AMWL

and DOT are exceptionally robust. Only for an error level of 20 cm the significance at My Thuan drops (i.e. 27 % of the disturbed time series at $p \leq 0.01$ and 67 % at $p \leq 0.001$), while the trend at Can Tho remains highly significant (99 % at $p \leq 0.001$) as given in Table 4. These results support public claims of higher and longer floods at Can Tho in the past decade. The opposite direction of the trends at Can Tho and My Thuan compared to those at Tan Chau and Chau Doc supports the hypothesis that high-dyke construction in the upper part of the VMD has altered the flood hazard downstream.

**3.2 Step change analysis and uncertainty analysis**

Figure 7 presents the results of the step-change analysis for the four main gauges in the VMD. We did not apply the Pettitt test for station Kratie, since flood changes in Kratie were negligible during the study period. The detected change points for both AMWL and DOT differ between upstream and downstream stations. The test identified step changes in 2005/2006 at the upstream stations Tan Chau and Chau Doc and in 2000/2001 at the stations Can Tho and My Thuan downstream of the high-

dyke areas.

AMWL at the upstream stations Tan Chau and Chau Doc decrease by roughly 10 %, but are not significant (both $p > 0.1$). The test on DOT also identifies a change point in 2005 for both stations, where the mean DOT decreases over 50%, but again with low significance (Tan Chau: $p = 0.14$; Chau Doc: $p = 0.12$). Arguably, this low significance can be attributed to the limited number of data points after the change point (ten data points). The test for robustness indicates that the detected step changes

are not robust against data errors, particularly for AMWL, as indicated by the markers in Fig. 7. Even assuming a small error of at most 5 cm, the timing of the step change can vary by about 20 years.





In contrast, the detected step changes in both AMWL and DOT for the downstream stations Can Tho and My Thuan are highly significant and robust. They amount to an increase in AMWL of 17 % at Can Tho and 12.5 % at My Thuan (both p ≤0.001). DOT is even three times longer in the post-2000 period compared to the pre-2000 period (p ≤ 0.001). The test for robustness indicates that the step change of DOT is almost always detected in the year 2000, even for an assumed error of up to 20 cm.

5   For AMWL the assumed errors can cause slight shifts of the detected step change in 2000, but the distribution of the MC-detected step changes are almost normally distributed around the median of 2000 (Fig. 7).

Dang et al. (2016) also reported a reduction of 21 % and 14 % (p > 0.05) in the annual mean water level at Tan Chau and Chau Doc, while for Can Tho they found that the water level increased by 27 % (p ≤ 0.05) between the pre- and post-2007 periods. However, we have to note the differences in the study designs. Dang et al. (2016) used the annual mean water level and set the

10  year 2007 as separation year based on the analysis of satellite images of inundation extents. Our study, in contrast, analyses flood indicators and derives the timing of step changes directly from the water level data. Hence, Dang et al. (2016) used the completion year of the majority of dyke systems, which were built during a number of years following the 2000 flood, whereas our approach rather detects the year where the construction starts to unfold its largest impact on the flood indicators. Thus it is likely that the different timing of the step changes refers to the same cause.

The facts that at both downstream stations step changes with identical timing are detected, that this timing corresponds to the start of the high-dyke construction in the upstream provinces An Giang and Dong Thap, and that these step changes are not detectable upstream of the high-dyke areas, suggest adverse impacts of upstream dyke development on downstream inundation dynamics. This suggestion is further analysed in the following sections presenting the hydraulic modelling results.

### 3.3 Impact of high-dyke development derived from hydraulic modelling

In the following the results of the hydrodynamic model for the two model setups, i.e. with and without high-dyke systems in An Giang and Dong Thap, are presented. Differences in inundation areas, depths and duration between the two model setups are given in Fig. 8, 9 and Table 5 for the two floods 2000 and 2011. Table S2 (supplement) summarizes the impact on inundation area for the nine flood-prone provinces in the VMD.

A first observation is that the high-dyke development leads to almost identical impacts for the two floods (e.g. compare the

upper and lower panel of Fig. 8). The maximum difference in terms of impact on inundation depth is 3.4 cm (station Tan Hiep, Table 5), whereas the differences in terms of impact on inundation duration are at most 1 day between the two floods. Although this similarity does not prove that the dyke system development has the same effect for other large flood events, it strongly suggests that the effect of dyke development is similar for large floods.

The results reveal that the high-dyke development increases inundation depth and duration at many locations in the VMD. The

impacts are not only found downstream of the high-dyke areas, e.g. at Can Tho or My Thuan, but can also be seen upstream of high-dyke areas, e.g. at Chau Doc or Tan Chau. This is a consequence of losing flood retention capacity in those areas protected by high-dyke and consequently backwater effects due to changes in in-channel flow. The impact varies from location to location (Table 5). It is rather minor for the two important floodplains in the VMD, i.e. POR and LXQ. An increase in maximum depth of ~5 cm and a change in inundated area of less than 2 % is found for the provinces Long An (POR) and Kien

Giang (LXQ), which are located farthest from the main rivers (Table S2). A significant reduction in inundation area of 32 % and 19 % is quantified for the provinces at An Giang and Dong Thap, respectively. The increase in AMWL of 5–6 cm at the two stations in these provinces (Chau Doc, Tan Chau) is far less than the negative slope detected (~15–17 mm year$^{-1}$) at these locations (Sec. 3.1.). This indicates that only a part of the detected weak downward trends at these locations could be explained by the construction of the high-dyke system.

High-dyke construction is found to have notable impact on water levels along the Mekong and Bassac Rivers downstream of the Vam Nao River connecting both rivers. Along the Mekong River, the impact ranges from ~12 cm at station Cao Lanh to ~8 cm at station My Thuan (Table 5). Along the Bassac River, the impact increases from station Vam Nao (~7 cm) through





Long Xuyen to Can Tho (~12 cm). Downstream of stations Can Tho and My Thuan, the impacts are smaller in the range of 3–7 cm.

Our results indicate that high-dyke development increases the duration of water levels exceeding the alarm level 1 by 17 days at Can Tho and 13 days at My Thuan (Table 5). This is expected to exacerbate urban flooding in Can Tho City with its very low topography and poor drainage system as reported in (Apel et al., 2016;Huong and Pathirana, 2013). In Can Tho City the total inundation area is not affected by the high-dyke construction, however, 6 % of the city area is converted from shallow inundation with depths below 1 m to deep inundation above 1 m (Table S2). The Vinh Long province, which is located between the Mekong and Bassac Rivers directly downstream of Can Tho and My Thuan, experiences large changes in inundation extent. Over 12 % of the total area of the province is altered from no/shallow inundation to deep inundation due to high-dyke development.

Notably, the impact on AMWL and DOT at the stations Can Tho and My Thuan is almost identical to the range estimated from the trend analysis. Applying the obtained trend slopes to the interval 2000−2011 yields an increase of 11−15 cm for AMWL, and 12−18 day for DOT. This agreement between the two independent methods enhances the credibility of the results.

### 3.4 Comparison of impacts driven by changes in the upper and lower boundaries and in the dyke system

In this section the relative impact of the three factors (i) changes in the upper boundary, (ii) high-dyke construction, and (iii) changes of the lower boundary, is evaluated by interchanging the upper boundary, the dyke system and the lower boundary of the model simulation for the reference years 2000 and 2011. The scenario set is given in Table 3 and the results are summarized in Table 5. The contributions of the three factors are compared to the baseline difference, which is defined as the difference in maximum water level of 2011 (scenario S3) and 2000 (scenario S1). The contributions were calculated as the mean of the two scenarios for each factor. The relative contributions are presented in Table 6 and Fig. 10. Table 5 reveals that for most stations the maximum water levels and the duration over threshold levels decrease when as upper boundary the flood hydrograph of 2000 is replaced with the higher volume flood of 2011. Interestingly, this is not the case for Can Tho and My Thuan. These stations are located in areas, where the magnitude of both river discharge and tidal water levels is important for AMWL and DOT during the annual flood events. Moreover, not only the magnitude, but also the temporal coincidence of maximum flood water levels and the spring tides is highly relevant. During the flood in 2000, the flood peak in Tan Chau and Chau Doc occurred on September 23[rd], coinciding with a neap tide period. On the contrary, the peak in 2011 occurred during the 1[st] week of October coinciding with a spring tide period. When the upper boundaries are interchanged, this timing is reversed: the peak flood levels of 2000 coincide with the neap tide in 2011, while the peaks of 2011 are combined with high tidal levels in 2000. This means that for the flood event of 2011 with the upper boundary of 2000 lower AMWL for Can Tho and My Thuan are simulated, although the peak discharge of the flood hydrograph of 2000 was lower than 2011. Analogously for the flood in 2000 with the upper boundary of 2011 higher AMWL are simulated due to the temporal coincidence of high flood water and spring tide condition. This effect causes positive differences in AMWL for these two locations, when the upper boundaries are interchanged. This effect also explains the comparatively large positive differences listed in Table 5 at these two stations when the lower boundaries are interchanged. For the DOT this effect is also observable for the scenario S5-S1, but for the analogous scenario the long period of high water levels in 2000 causes higher DOT, even if the period of high water levels do not coincide with the spring tide period.

A comparison of the changes in AMWL and DOT in Table 5 reveals a spatial pattern: The upstream boundary has the largest impact on the stations located north of Long Xuyen and Cao Lanh, and at stations with larger distances from the main rivers. In these areas the tidal amplitude is attenuated due to the large distance from the sea and/or the length of the channel network.

The situation at the stations downstream of the high-dyke areas and along the main rivers (stations Long Xuyen, Cao Lanh, Can Tho, My Thuan) is more complex. Here the changes caused by the different factors have similar magnitudes, whereas the





tidal influence increases in downstream direction. Hence, the factors can partly compensate each other, and an attribution of observed changes to individual factors requires a careful inspection.

Table 6 and Figure 10 present the contributions of the three factors in relation to the observed differences between the floods in 2000 and 2011. Apparently, the sum of the single contributions does not equal the observed differences. This is a consequence of non-linear interactions, including, for example, the problem of timing between flood peaks and tidal characteristics. The magnitude of the discrepancy varies widely between the stations from a close match to almost 50 %. Most of the stations with small discrepancies are located farthest north and farthest away from the main rivers. For those stations the upstream boundary causes changes of 120–130 % of the baseline difference. Dyke construction and, to a lesser extent, the change in the lower boundary counteract these changes.

Downstream of Vam Nao the contribution of the upper boundary diminishes to only 20–50 % of the baseline difference. In this region the other two factors have higher importance, but the relative contribution changes from station to station. With respect to the motivation of the study, i.e. the quantification of the impact of high-dyke construction on flood water levels, this region is the most affected by the high-dyke development. This factor amounts to 65–80 % of the baseline difference. Further downstream, i.e. at Can Tho and My Thuan, the lower boundary becomes the prevailing factor, amounting to more than 80 % of the baseline difference. The water level changes caused by the high-dyke system is reduced to about 30 %.

These findings disagree with public and several officials' claims that the high-dyke development in the upper part of the MD is the main factor for the higher and longer floods in Can Tho and the central part of the Mekong Delta. For this region it is rather the combination of all three factors, with the lower boundary, respectively the timing of high flood levels and tidal levels, dominating. The further downstream the station location, the higher is the impact of the tidal water level. Hence, sea level rise and land subsidence are important factors for future inundation dynamics in the central and coastal areas of the VMD. These findings are in line with Fujihara et al. (2015) who identified sea level rise and land subsidence as the main factors controlling alteration of minimum and maximum water levels in the middle part and coastal zones of the VMD.

## 4 Conclusions

The present study sheds light on the link between changes in flood dynamics in the VMD and high-dyke construction in the Northern provinces An Giang and Dong Thap. The research was motivated by the recent discussions in the Vietnamese public, and by the media and officials regarding the role of the high-dyke development. These discussions were triggered by flood in 2011, where higher flood levels occurred in the middle part of the VMD compared to the flood in 2000, which had a much larger flood volume and a longer duration of high water levels, but lower peak discharges. However, the widespread claim that high-dyke development is the sole cause for increased flood water levels has not been investigated in detail. Particularly, the effects of the possible causes – different flood hydrograph characteristics, high-dyke construction, and different tidal dynamics – have not been quantified to date.

In a first step we performed a trend analysis of indicators of flood severity, i.e. annual maximum water levels (AMWL) and the duration of water levels above a warning threshold (DOT) for the period 1978–2015. Negative trends of low significance were found for the upper part of the delta (stations Chau Doc, Tan Chau). On the contrary, strongly increasing and highly significant trends were detected downstream of the areas with the large-scale high-dyke development. These trends were also highly robust against measurement errors. The Pettitt test revealed a step change at the stations downstream of the high-dyke areas around the year 2000 with high significance and robustness. The timing of the step change coincides with the initiation of the high-dyke development. This result differs from the step change identified by Dang et al. (2016) at the year 2006. However, this difference is a consequence of different methodologies; our statistical approach identified the start of the high-dyke construction period, while Dang et al. (2016) defined the end of the main construction period as step change date. Thus, our analyses are in line with the widespread public claim about the adverse impacts of the high-dyke construction on





downstream water levels. Our trend analyses also strengthen the conclusions of the earlier, less refined, studies of Fujihara et al. (2015) and Dang et al. (2016).

However, trend analyses do not allow to separate the contribution of high-dyke development from the other important factors, i.e. the severity of the flood entering the MD, and the tidal influence. This separation was achieved by running an up-to-date

hydrodynamic flood model of the whole MD for a set of scenarios with interchanged boundaries and dyke system state. The model simulations indicated that at Can Tho and My Thuan in the central VMD an increase of 9–13 cm in maximum water level and of 15 day in duration above the 1st official warning level can be attributed to the high-dyke development. However, this explains only about 30 % of the observed differences between the floods in 2011 and 2000.

The hydraulic model scenarios also demonstrated the importance of the different boundary conditions and how this importance

varies in space. In the northern part of the VMD the flood hydrograph entering the MD has the highest importance, while further downstream the tidal influence dominates. The simulations further revealed that not only the tidal level, but also the timing of spring tides in relation to maximum flood water level plays an important role for the flood hazard in central and coastal VMD. For central VMD (Can Tho and My Thuan) it was found that the most dominant factor was the tidal impact. The isolated tidal impact amounts to about 80% of the observed differences between the floods in 2011 and 2000. The higher

tidal level of 2011 and the coincidence of spring tide and high flood levels caused differences of about +19 cm and +32 cm at My Thuan and Can Tho, respectively, whereas the upper boundary caused an increase of 7–8 cm. Thus, the claims that the dyke development has altered the flood hazard in the areas downstream can be partially confirmed, but not the claim that it is the only cause. In fact, for central VMD the lower boundary has a 2–3 times higher influence.

For the flood in 2011 the coincidence of a late flood peak in October with a spring tide period was the main cause for the

20 exceptional flooding in My Thuan and particularly Can Tho. Hence, flood risk management plans should consider changes in the lower boundary and the timing of high flood flows and spring tides. A clear implication of our research is that higher flood levels as usual have to be expected in central and coastal VMD if flood water levels peak in October, which is the period of spring tides in the East Sea. This insight should be considered when flood warnings are issued.

Another important implication resulting from the link between inundation dynamics and lower boundary is that flood levels in

central and coastal VMD have to be expected to increase in the coming decades. Sea levels in the East Sea surrounding the MD rose by 3.1 mm year⁻¹ at My Thanh and 3.5 mm year⁻¹ at Vung Tau during 1985–2010 (Hak et al. (2016), and sea level rise is expected to continue (IPCC (2014). To make the situation even more alarming, several cities in the VMD suffer from local subsidence due to over-exploitation of groundwater, for instance, 20 mm year⁻¹ for Can Tho during 2006–2010 (Erban et al., 2014). This development adds to the climate driven sea level rise and will cause even higher effective tidal water levels.

Against this background, it is worth noting that the high-dyke system has the potential to be harnessed for flood mitigation in central VMD. Including the operation and flooding of the floodplains in Dong Thap and An Giang in flood management plans on a delta-wide organizational level would reduce the flood hazard in central VMD by re-using the natural floodplain storage. This could be organized for individual flood events, if large floods, and particularly flood peaks in October, are forecasted. Another option would be a long-term plan for counteracting the effective sea level rise by partial flooding of the floodplains

protected by high-dyke on an (multi-)annual rotation basis. Such management options would require a close cooperation and coordination between the provinces and districts, possibly overseen by national agencies. Additionally, compensation schemes would be required for farmers affected by emergency flooding of their fields where a summer crop has already been planted. Finally, the updated quasi-2-dimensional flood model performs better than the previous versions and proved to be a valuable tool for understanding and quantifying temporal changes in flood characteristics in this highly complex delta. The model can

be used to investigate the impacts of hydropower dam development, climate change and water management, such as the very likely expansion of high-dyke areas in the VMD, to delta inundation.





**Acknowledgement**

This study is a part of the first author PhD work, which is funded by the German Academic Exchange Service (DAAD). We acknowledge Dr. Akihiko Kotera from Kobe University for providing processed MODIS images. The authors want to specially thank Dr. Nguyen Nghia Hung, Dr. To Quang Toan, Mr. Pham The Vinh and various colleagues at SIWRR for their support

in providing dyke survey and DEM data, together with numerous recommendations and comments on the data and modelling.

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





**Table 1**. Model performance: location, parameter and results for the two flood events in 2000 and 2011. Model performance is quantified by the Nash-Sutcliffe efficiency (NSE).

| No. | Station | Location | Flood event in 2000 | | Flood event in 2011 | |
|---|---|---|---|---|---|---|
| | | | *H* | *Q* | *H* | *Q* |
| 1 | Kampong Cham | CFP | 0.88 | | 0.84 | |
| 2 | Prek Kdam | CFP | 0.95 | | 0.96 | |
| 3 | Phnom Penh (port) | CFP | 0.93 | | 0.96 | |
| 4 | Neakluong | CFP | 0.98 | | 0.98 | |
| 5 | KhoKhel | CFP | *N/A** | | 0.86 | |
| 6 | Tan Chau | VMD | 0.96 | *N/A* | 0.97 | 0.80 |
| 7 | Chau Doc | VMD | 0.97 | 0.76 | 0.87 | 0.96 |
| 8 | Long Xuyen | VMD | 0.97 | | 0.96 | |
| 9 | Cao Lanh | VMD | 0.96 | | 0.96 | |
| 10 | Vam Nao | VMD | 0.98 | 0.74 | 0.96 | 0.74 |
| 11 | Can Tho | VMD | 0.76 | 0.73 | 0.86 | 0.74 |
| 12 | My Thuan | VMD | 0.72 | 0.68 | 0.80 | 0.73 |
| 13 | Xuan To | VMD | 0.88 | | 0.89 | |
| 14 | Tri Ton | VMD | 0.89 | | 0.94 | |
| 15 | Moc Hoa | VMD | 0.78 | | 0.94 | |
| 16 | Hung Thanh | VMD | 0.74 | | 0.92 | |
| 17 | Kien Binh | VMD | 0.91 | | 0.80 | |
| 18 | Tan Hiep | VMD | *N/A* | | 0.80 | |
| 19 | Vi Thanh | VMD | *N/A* | | 0.78 | |

[*] *N/A means no data for model performance evaluation*

**Table 2**. Comparison of model performance for the flood event of 2011 between the pre- and post-updated versions. Values show the mean of all stations, resp. satellite images used for the model performance evaluation.

| Objectives | Pre-updated version* | | Post-updated version | |
|---|---|---|---|---|
| | NSE | FAI | NSE | FAI |
| Water level (m) | 0.84 | | 0.90 | |
| Discharge (m³s⁻¹) | 0.63 | | 0.79 | |
| Inundation (flood area index FAI) | | 0.46 | | 0.64 |

* from Manh et al. (2014)





**Table 3.** Scenarios used to separate the impacts of high-dyke development, and changes in upstream and lower boundary conditions to the alteration of inundation dynamics in the VMD. Scenarios 1 and 3 are the baseline scenarios.

| Scenario name | upper boundaries | dyke development | downstream boundaries |
|---|---|---|---|
| S1: u00-nHD-d00 | 2000 | no high-dyke | 2000 |
| S2: u00-yHD-d00 | 2000 | with high-dyke | 2000 |
| S3: u11-yHD-d11 | 2011 | with high-dyke | 2011 |
| S4: u11-nHD-d11 | 2011 | no high-dyke | 2011 |
| S5: u11-nHD-d00 | 2011 | no high-dyke | 2000 |
| S6: u00-yHD-d11 | 2000 | with high-dyke | 2011 |
| S7: u00-nHD-d11 | 2000 | no high-dyke | 2011 |
| S8:u11-yHD-d00 | 2011 | with high-dyke | 2000 |
| S2 – S1 | impacts of high-dyke development (setup1 – flood event 2000) | | |
| S3 – S4 | impacts of high-dyke development (setup2 – flood event 2011) | | |
| S5 – S1 | impacts of upper boundaries (setup1 – flood event 2000) | | |
| S3 – S6 | impacts of upper boundaries (setup2 – flood event 2011) | | |
| S7 – S1 | impacts of downstream boundaries (setup1 – flood event 2000) | | |
| S3 – S8 | impacts of downstream boundaries (setup2 – flood event 2011) | | |





**Table 4.** Uncertainty analysis of detected trends obtained from Mann – Kendall's test and Sen' slope to evaluate the trend robustness against data errors ($C_v$ denotes the Coefficient of Variation).

| Station | test parameter | error scale | Mann-Kendall test statistic Z value | | | | significance level of 1000 disturbed time-series* | | | | | | Sen's slope estimation | | | |
|---|---|---|---|---|---|---|---|---|---|---|---|---|---|---|---|---|
| | | | 2.5% quantile | 97.5% quantile | median | $C_v$ | no trend | $p > 0.1$ | $p \leq 0.1$ | $p \leq 0.05$ | $p \leq 0.01$ | $p \leq 0.001$ | 2.5% quantile | 97.5% quantile | median | $C_v$ |
| Kratie | AMWL | ± 5 cm | -0.03 | 0.18 | 0.08 | 0.71 | 129 | 871 | | | | | -0.6 | 3.4 | 1.3 | 0.77 |
| | | ± 10 cm | -0.03 | 0.23 | 0.08 | 0.95 | 118 | 882 | | | | | -1.0 | 5.4 | 2.2 | 0.74 |
| | | ± 20 cm | -0.13 | 0.35 | 0.10 | 1.22 | 107 | 893 | | | | | -2.6 | 7.4 | 2.3 | 1.11 |
| | AFV | ± 1% | -0.45 | -0.23 | -0.33 | 0.19 | | 1000 | | | | | -0.6 | -0.3 | -0.4 | 0.14 |
| | | ± 3% | -0.68 | -0.18 | -0.43 | 0.30 | | 1000 | | | | | -0.7 | -0.2 | -0.5 | 0.26 |
| | | ± 6% | -0.93 | -0.08 | -0.50 | 0.43 | 8 | 992 | | | | | -0.9 | -0.1 | -0.5 | 0.41 |
| Tan Chau | AMWL | ± 5 cm | -2.11 | -1.73 | -1.94 | 0.05 | | 3 | 565 | 432 | | | -18.4 | -15.2 | -16.8 | 0.05 |
| | | ± 10 cm | -2.19 | -1.56 | -1.89 | 0.08 | | 68 | 574 | 358 | | | -19.2 | -14.2 | -16.9 | 0.07 |
| | | ± 20 cm | -2.34 | -1.38 | -1.84 | 0.14 | | 214 | 444 | 342 | | | -21.2 | -12.8 | -16.9 | 0.13 |
| | DOT | ± 5 cm | -2.58 | -2.36 | -2.48 | 0.02 | | | | 961 | 39 | | -2.6 | -2.4 | -2.5 | 0.02 |
| | | ± 10 cm | -2.54 | -2.26 | -2.40 | 0.03 | | | | 993 | 7 | | -2.5 | -2.3 | -2.4 | 0.03 |
| | | ± 20 cm | -2.51 | -2.20 | -2.36 | 0.03 | | | | 995 | 5 | | -2.5 | -2.2 | -2.4 | 0.03 |
| Chau Doc | AMWL | ± 5 cm | -1.73 | -1.41 | -1.58 | 0.05 | | 816 | 184 | | | | -16.4 | -13.4 | -15.0 | 0.05 |
| | | ± 10 cm | -1.81 | -1.31 | -1.58 | 0.08 | | 712 | 287 | 1 | | | -17.4 | -12.5 | -14.9 | 0.09 |
| | | ± 20 cm | -1.91 | -1.11 | -1.52 | 0.14 | | 707 | 272 | 21 | | | -19.2 | -10.8 | -14.9 | 0.14 |
| | DOT | ± 5 cm | -1.65 | -1.43 | -1.53 | 0.03 | | 971 | 29 | | | | -1.6 | -1.4 | -1.5 | 0.03 |
| | | ± 10 cm | -1.65 | -1.38 | -1.52 | 0.04 | | 963 | 37 | | | | -1.6 | -1.4 | -1.5 | 0.04 |
| | | ± 20 cm | -1.62 | -1.27 | -1.46 | 0.06 | | 984 | 16 | | | | -1.6 | -1.3 | -1.5 | 0.06 |
| Can Tho | AMWL | ± 5 cm | 5.18 | 6.03 | 5.61 | 0.04 | | | | | | 1000 | 11.9 | 13.8 | 12.8 | 0.04 |
| | | ± 10 cm | 4.73 | 6.06 | 5.36 | 0.06 | | | | | | 1000 | 10.9 | 14.7 | 12.9 | 0.08 |
| | | ± 20 cm | 3.49 | 5.48 | 4.50 | 0.11 | | | | | 11 | 989 | 9.2 | 16.4 | 12.8 | 0.15 |
| | DOT | ± 5 cm | 4.84 | 5.32 | 5.09 | 0.02 | | | | | | 1000 | 4.8 | 5.3 | 5.1 | 0.02 |
| | | ± 10 cm | 4.76 | 5.42 | 5.09 | 0.03 | | | | | | 1000 | 4.8 | 5.4 | 5.1 | 0.03 |
| | | ± 20 cm | 4.74 | 5.71 | 5.23 | 0.05 | | | | | | 1000 | 4.7 | 5.7 | 5.2 | 0.05 |
| My Thuan | AMWL | ± 5 cm | 4.45 | 5.56 | 5.03 | 0.05 | | | | | | 1000 | 8.0 | 10.2 | 9.1 | 0.06 |
| | | ± 10 cm | 3.87 | 5.41 | 4.63 | 0.09 | | | | | 1 | 999 | 7.3 | 11.2 | 9.1 | 0.11 |
| | | ± 20 cm | 2.21 | 4.78 | 3.58 | 0.18 | | | 5 | 52 | 271 | 672 | 5.4 | 12.8 | 9.2 | 0.20 |
| | DOT | ± 5 cm | 4.20 | 4.71 | 4.46 | 0.03 | | | | | | 1000 | 4.2 | 4.7 | 4.5 | 0.03 |
| | | ± 10 cm | 4.04 | 4.75 | 4.39 | 0.04 | | | | | | 1000 | 4.0 | 4.8 | 4.4 | 0.04 |
| | | ± 20 cm | 3.68 | 4.66 | 4.17 | 0.06 | | | | | | 1000 | 3.7 | 4.7 | 4.2 | 0.06 |

*number of cases when the detected trend is significant at certain level within 1000 disturbed time series





**Table 5.** Contribution of each of the three factors, i.e. upper boundary, high-dyke construction and lower boundary, to the alteration of maximum water level (in cm) and duration over threshold (in days) at key gauging stations in the VMD.

| station code | gauge station [1] | change in maximum water level | | | | | | | | | change in duration over threshold [2] | | | | | | | | |
| | | upstream boundary [3] | | | high-dyke | | | downstream boundary | | | upstream boundary | | | high-dyke | | | downstream boundary | | |
| | | S5-S1 | S3-S6 | mean | S2-S1 | S3-S4 | mean | S7-S1 | S3-S8 | mean | S5-S1 | S3-S6 | mean | S2-S1 | S3-S4 | mean | S7-S1 | S3-S8 | mean |
|---|---|---|---|---|---|---|---|---|---|---|---|---|---|---|---|---|---|---|---|
| 8 | Chau Doc | -27.6 | -24.8 | -26.0 | 5.0 | 4.2 | 4.6 | -0.3 | 2.9 | 1.3 | -18.0 | -16.0 | -17.0 | 0.0 | 0.0 | 0.0 | 0.0 | 2.0 | 1.0 |
| 7 | Tan Chau | -28.9 | -26.0 | -28.0 | 6.0 | 4.9 | 5.5 | -1.0 | 1.9 | 0.5 | -19.0 | -18.0 | -19.0 | 1.0 | 0.0 | 1.0 | 0.0 | 1.0 | 1.0 |
| 9 | Vam Nao | -18.4 | -23.4 | -21.0 | 6.9 | 6.9 | 6.9 | 5.0 | 1.0 | 3.0 | | | | | | | | | |
| 10 | Long Xuyen | -8.2 | -8.7 | -8.5 | 10.1 | 12.1 | 11.1 | 13.0 | 12.8 | 12.9 | | | | | | | | | |
| 11 | Cao Lanh | -0.5 | -5.4 | -2.9 | 12.1 | 11.1 | 11.6 | 13.9 | 9.1 | 11.5 | | | | | | | | | |
| 12 | Can Tho | 10.2 | 4.0 | 7.1 | 13.5 | 11.9 | 12.7 | 35.7 | 29.8 | 32.8 | 12.0 | -5.0 | 4.0 | 18.0 | 18.0 | 18.0 | 38.0 | 21.0 | 30.0 |
| 13 | My Thuan | 12.8 | 3.5 | 8.1 | 8.1 | 9.0 | 8.5 | 23.7 | 14.1 | 18.9 | 13.0 | -4.0 | 5.0 | 13.0 | 12.0 | 13.0 | 24.0 | 6.0 | 15.0 |
| 16 | Moc Hoa | -31.8 | -31.1 | -32.0 | 5.3 | 5.0 | 5.1 | 0.7 | 2.2 | 1.5 | | | | | | | | | |
| 14 | Hung Thanh | -24.5 | -25.2 | -25.0 | 4.0 | 3.5 | 3.7 | 1.6 | 0.8 | 1.2 | | | | | | | | | |
| 15 | Kien Binh | -17.2 | -12.4 | -15.0 | 5.0 | 4.8 | 4.9 | 1.1 | 6.0 | 3.6 | | | | | | | | | |
| 17 | Xuan To | -27.6 | -24.8 | -26.0 | 4.3 | 3.3 | 3.8 | -0.6 | 3.3 | 1.3 | | | | | | | | | |
| 18 | Tri Ton | -15.5 | -18.4 | -17.0 | 7.3 | 6.0 | 6.7 | 2.7 | 0.7 | 1.7 | | | | | | | | | |
| 19 | Tan Hiep | -5.7 | -1.1 | -3.4 | 5.9 | 2.5 | 4.2 | 2.7 | 9.0 | 5.9 | | | | | | | | | |
| 20 | Vi Thanh | -15.8 | -11.1 | -14.0 | 3.1 | 3.6 | 3.4 | 20.0 | 24.4 | 22.2 | | | | | | | | | |

[1] Stations are listed from upstream to downstream and from small to large distance from main rivers; station locations are given Fig.1 and Table S1.

[2] Calculated for 4 stations where trend tests were performed.

[3] $S5 - S1$: (u11nHDd00) − (u00nHDd00); $S3 - S6$: (u11yHDd11) − (u00yHDd11)

$S2 - S1$: (u00yHDd00) − (u00nHDd00); $S3 - S4$: (u11yHDd11) − (u11nHDd11)

$S7 - S1$: (u00nHDd11) − (u00nHDd00); $S3 - S8$: (u11yHDd11) − (u11yHDd00)



**Table 6.** Contribution of each factor to the alteration of water level between the floods 2000 and 2011 for key gauge stations in the VMD. The stations are listed from upstream to downstream and from small to large distance to the main rivers to remote. Station numbers refer to Fig. 1.

| station code | gauge station | base-line (S3 - S1) | upper bound. | high-dyke | lower bound. | summed changes | upper bound. | high-dyke | lower bound. | summed changes |
|---|---|---|---|---|---|---|---|---|---|---|
| | | [cm] | | | | [cm] | in percentage of baseline | | | % baseline |
| 8 | Chau Doc | -19.6 | -26.2 | 4.6 | 1.3 | -20.3 | -133.7 | 23.5 | 6.6 | 103.6 |
| 7 | Tan Chau | -20.6 | -27.5 | 5.5 | 0.5 | -21.5 | -133.5 | 26.7 | 2.4 | 104.4 |
| 9 | Vam Nao | -10.4 | -20.9 | 6.9 | 3.0 | -11 | -201.0 | 66.3 | 28.8 | 105.8 |
| 10 | Long Xuyen | 17.1 | -8.5 | 11.1 | 12.9 | 15.5 | -49.7 | 64.9 | 75.4 | 90.6 |
| 11 | Cao Lanh | 14.1 | -2.9 | 11.6 | 11.5 | 20.2 | -20.6 | 82.3 | 81.6 | 143.3 |
| 12 | Can Tho | 40.1 | 7.1 | 12.7 | 32.8 | 52.6 | 17.7 | 31.7 | 81.8 | 131.2 |
| 13 | My Thuan | 28 | 8.1 | 8.5 | 18.9 | 35.5 | 28.9 | 30.4 | 67.5 | 126.8 |
| 16 | Moc Hoa | -25.1 | -31.5 | 5.1 | 1.5 | -24.9 | -125.5 | 20.3 | 6.0 | 99.2 |
| 14 | Hung Thanh | -19.7 | -24.9 | 3.7 | 1.2 | -20 | -126.4 | 18.8 | 6.1 | 101.5 |
| 15 | Kien Binh | -7.8 | -14.8 | 4.9 | 3.6 | -6.3 | -189.7 | 62.8 | 46.2 | 80.8 |
| 17 | Xuan To | -20.5 | -26.2 | 3.8 | 1.3 | -21.1 | -127.8 | 18.5 | 6.3 | 102.9 |
| 18 | Tri Ton | -7.1 | -17.0 | 6.7 | 1.7 | -8.6 | -239.4 | 94.4 | 23.9 | 121.1 |
| 19 | Tan Hiep | 11.6 | -3.4 | 4.2 | 5.9 | 6.7 | -29.3 | 36.2 | 50.9 | 57.8 |
| 20 | Vi Thanh | 16.2 | -13.5 | 3.4 | 22.2 | 12.1 | -83.3 | 21.0 | 137.0 | 74.7 |

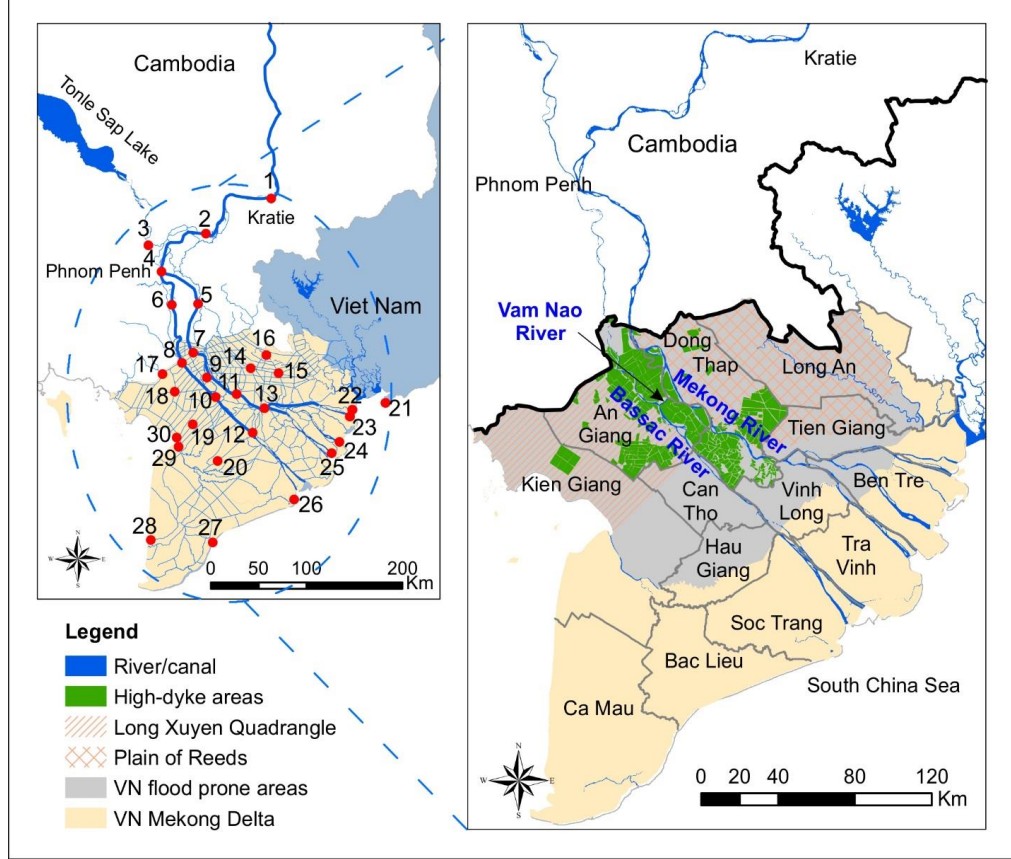

**Figure 1.** The Vietnamese Mekong Delta, its flood prone areas and location of measuring stations (red dots). The names in black indicate the provinces in the VMD.





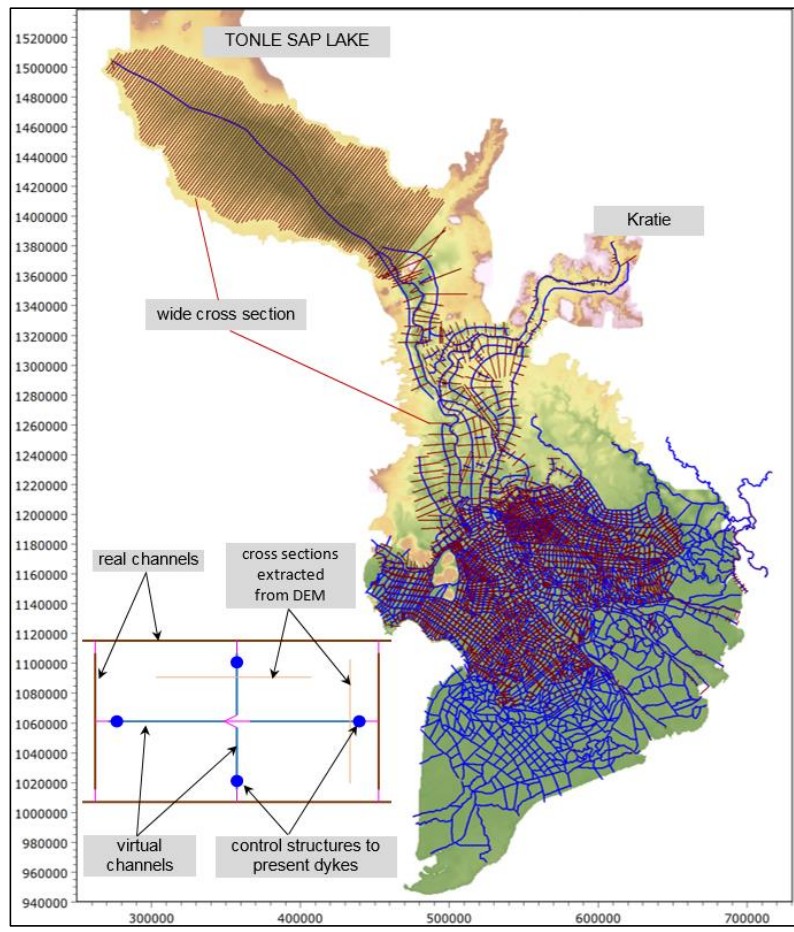

**Figure 2.** The channel network of the quasi-2D flood model for the MD, and the simulation concept for flood compartments in the model.

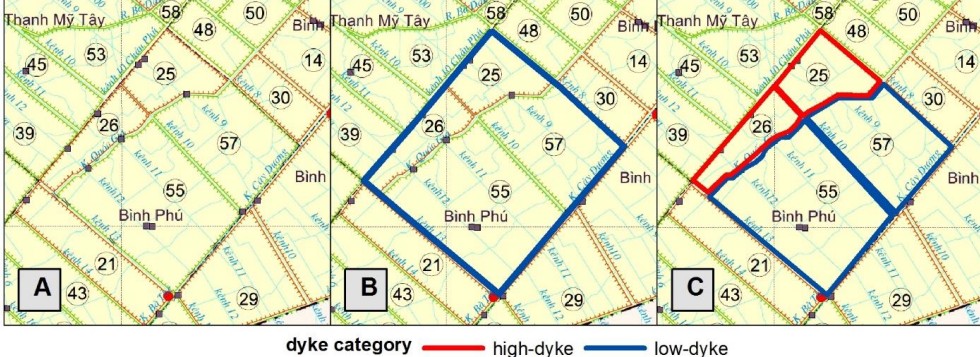

**Figure 3.** An example of a flood compartment updated in the presented model: **(a)** dyke survey map: orange lines mean high-dyke, green lines represent low-dyke, **(b)** presented as a single compartment in the original model, **(c)** updated model river network and dyke system.





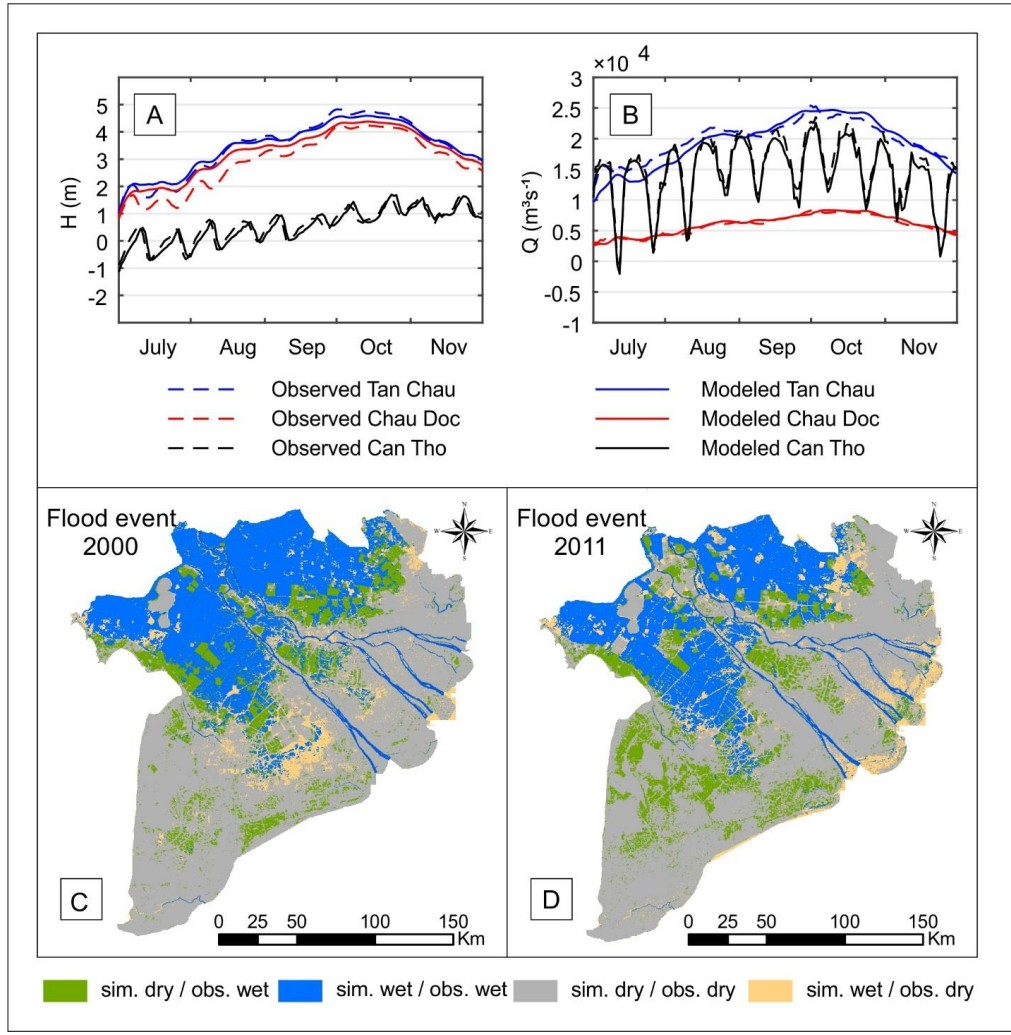

**Figure 4.** Model performance: comparison of gauged and simulated water level **(a)** and discharge **(b)** at stations Tan Chau, Chau Doc and Can Tho for the 2011 flood; comparison of observed i.e. derived from satellite data and simulated maximum inundation extent for the floods in 2000 **(c)** and 2011 **(d)**. Grey and blue color indicate agreement, green color means observed inundation but no inundation simulated, and yellow implies simulated inundation but no inundation observed.



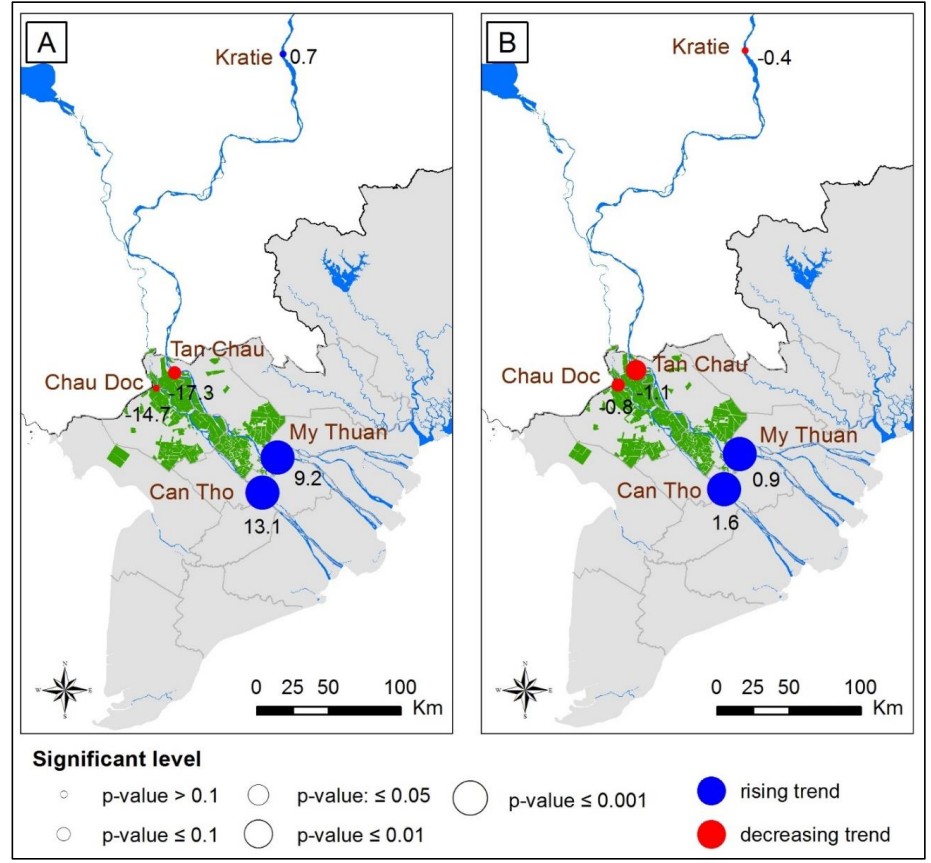

**Figure 5.** Flood trends at key locations during the study period 1978 – 2015. (A) Trend of annual maximum water level, slope presented in mm year$^{-1}$. (B) Trend of duration of peak over threshold and annual flood volume, slope presented in day year$^{-1}$ (DOT) and in km³ year$^{-1}$ (flood volume). Blue circles indicate rising trends and red circles imply decreasing trends. Circle size is proportional to significance level. The numbers present estimated slope. High-dyke areas are marked in green.

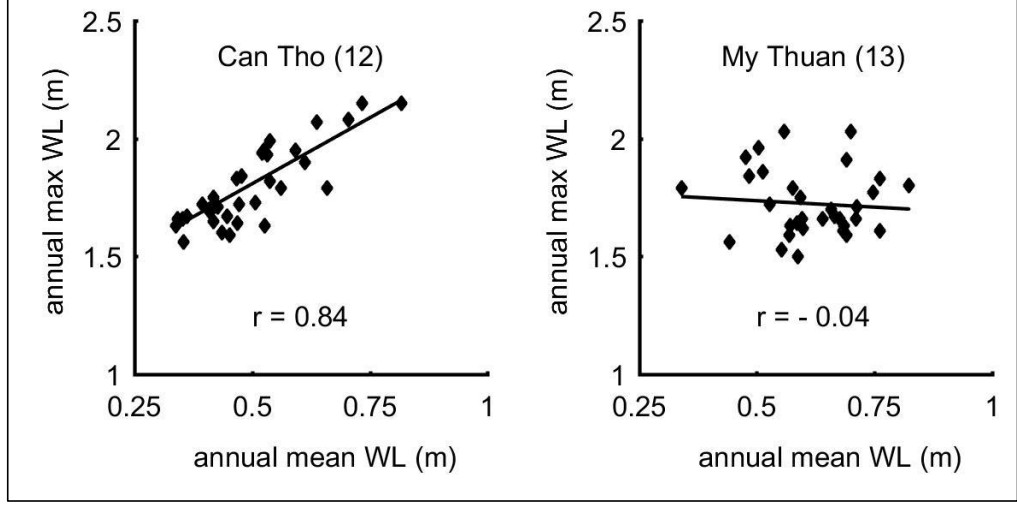

**Figure 6.** Strong correlation between annual maximum and mean water level at Can Tho, but not correlate at My Thuan, data from 1978–2015. The numbers next to the station names refer to the numbers in Fig.1.





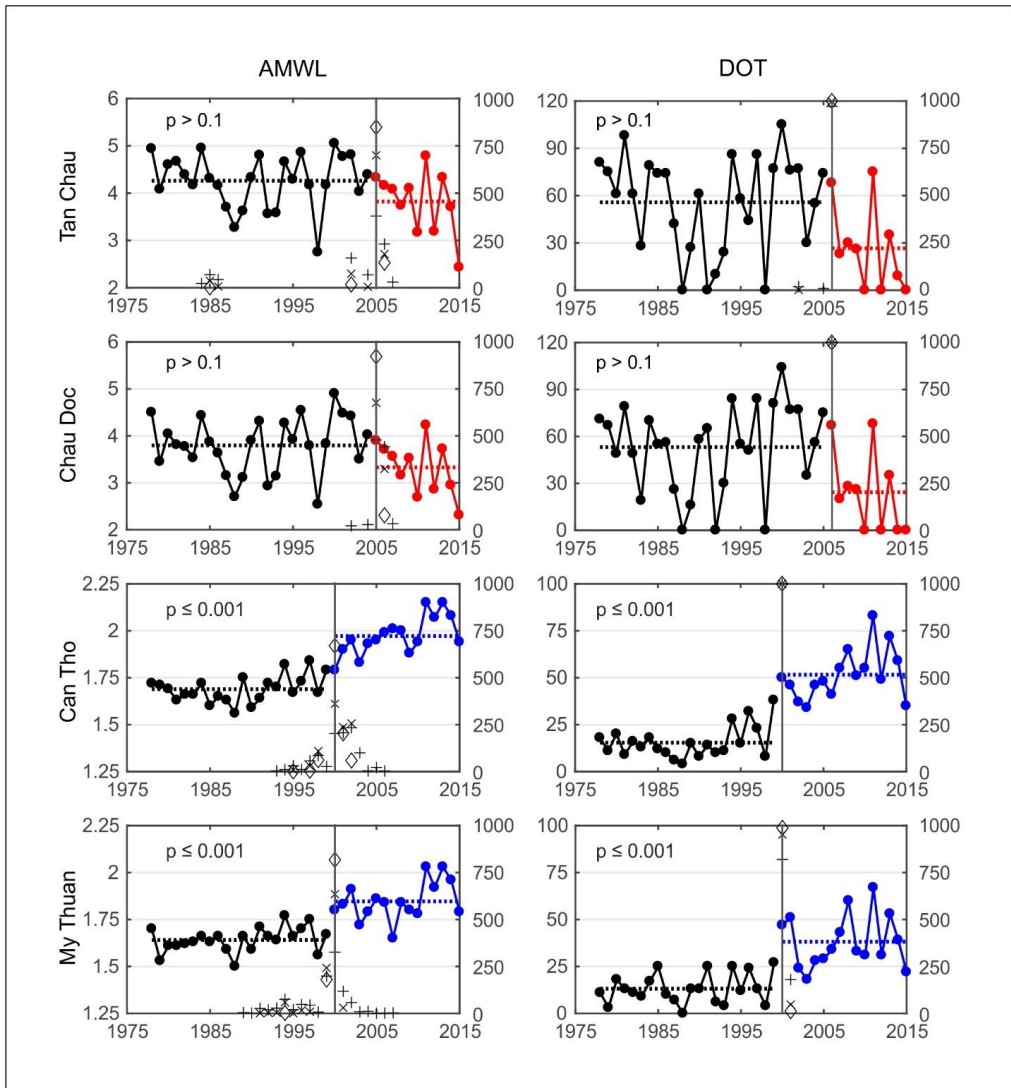

**Figure 7.** Step-change analysis including the uncertainty analysis at four stations in the VMD. Left column presents results for annual maximum water level (in meters) and the right column for flood duration over thresholds (in days). Data before the change point is presented in black. The second period is presented in red for negative step change and in blue for a positive change. Dashed lines are mean values of partial datasets. Results of the uncertainty analysis are presented using the second vertical axis on the right, as the number of times that a step change occurred in a certain year at three error levels (5%: diamond, 10%: cross, 15%: plus).





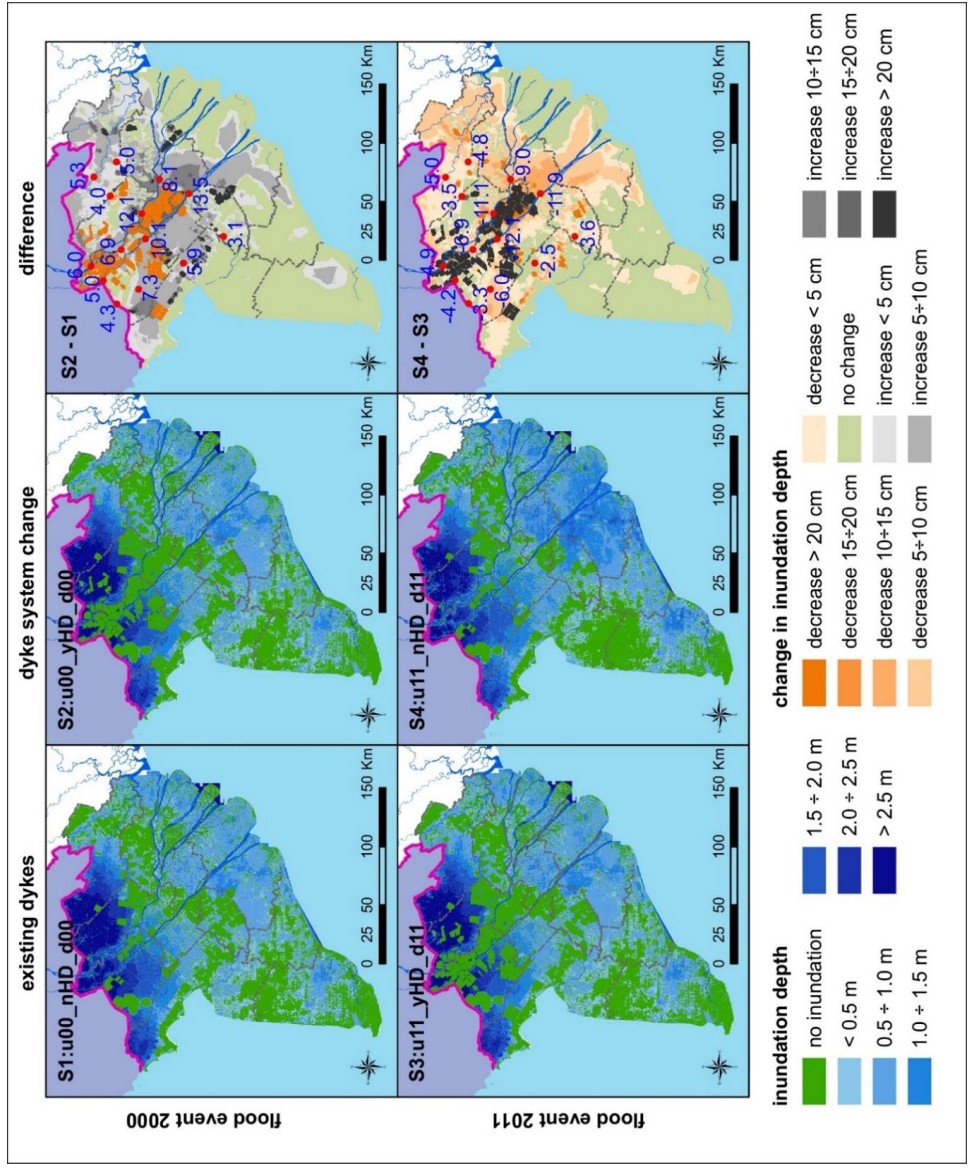

**Figure 8.** Impacts of upstream high-dyke development on downstream inundation area and depth for the flood event 2000 (upper row) and the flood 2011 (lower row). Left column: dykes as existing in the respective year. Middle column: dyke system interchanged between the two years. Right column: difference between the two dyke scenarios.



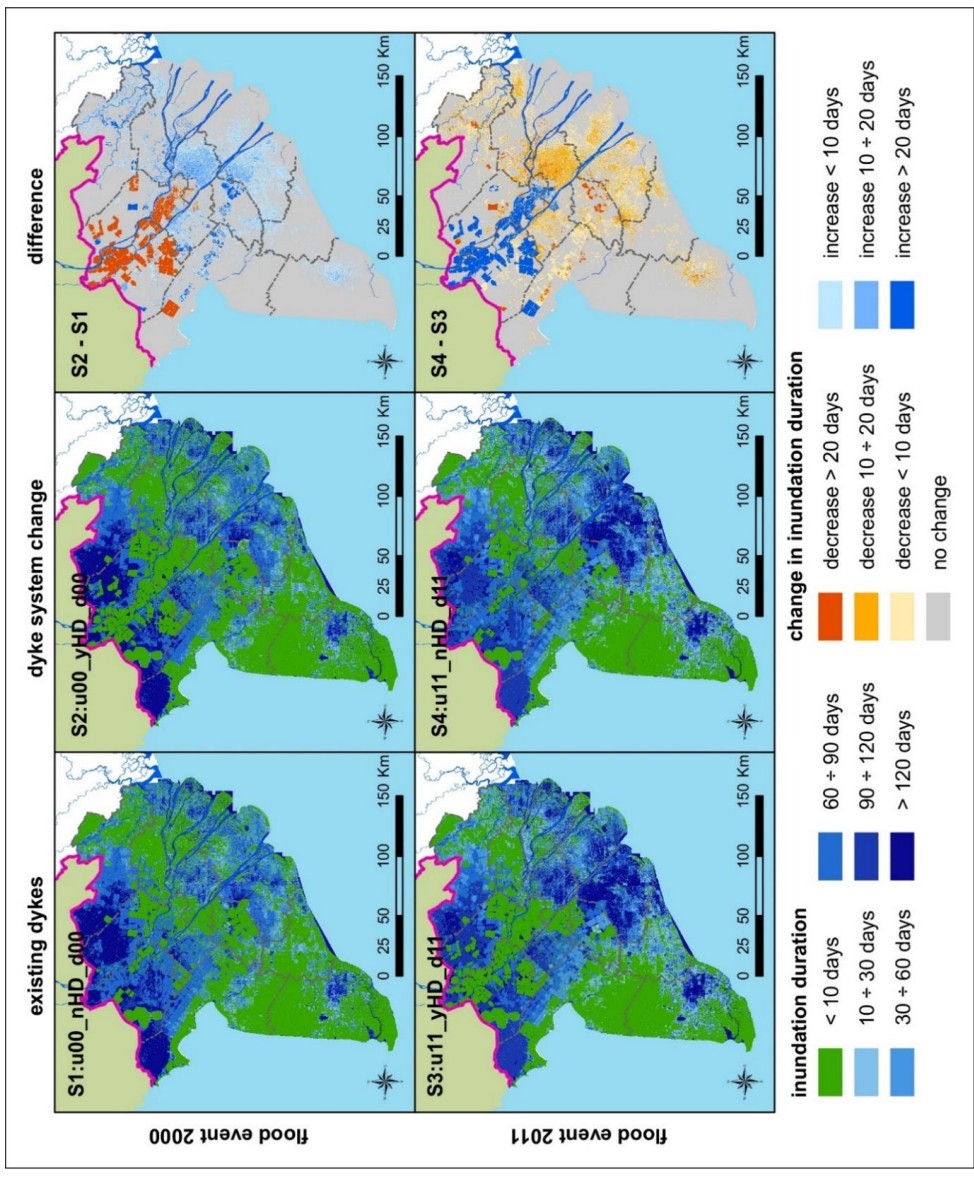

**Figure 9.** Impacts of upstream high-dyke development on downstream inundation duration for the flood event 2000 (upper row) and the flood 2011 (lower row). Left column: dykes as existing in the respective year. Middle column: dyke system interchanged between the two years. Right column: difference between the two dyke scenarios.





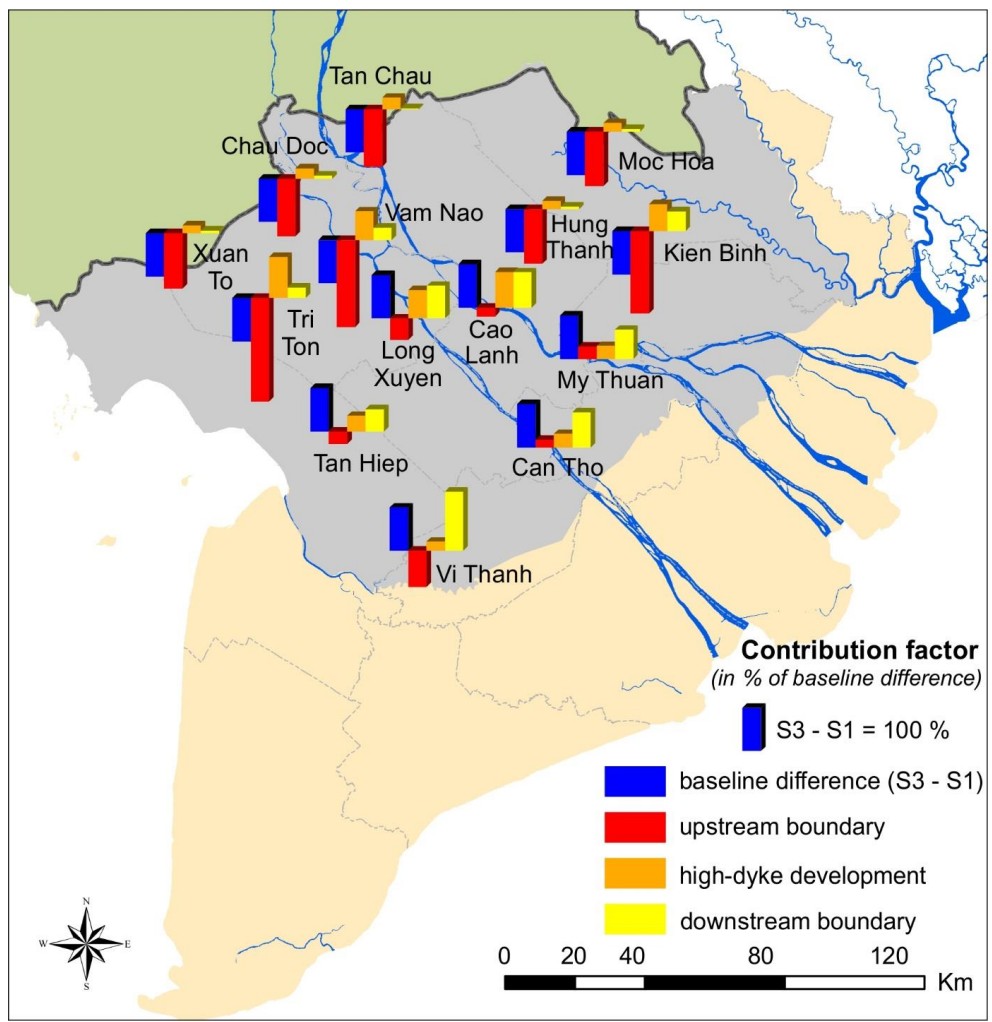

**Figure 10.** Contribution of each factor to the alteration of water levels between the floods 2000 and 2011 for key gauge stations in the VMD. The difference between the two scenarios S3 and S1 is considered to be 100%.