# Peer review of "Has dyke development in the Vietnamese Mekong Delta shifted flood hazard downstream?"

_Hydrology and Earth System Sciences, 2017_

## Referee Comment (RC1) · Anonymous Referee #1 · 18 May 2017

The paper presents the interesting case of the flooding in the Vietnamese delta and the consequences of dyke construction on the flooding pattern. The subject well presented and the approach is detailed. Though as authors are mentioning there should be further studies about the possible causes for the new flooding patterns, the paper presents in depth analysis of the different flooded areas in the downstream of the catchment where large-scale high-dyke development were performed. I did miss however details of the quasi-2D model used. Can you please give more explanations on the model, not to repeat what is in the reference, but to explain the underlying equations and the solution of them. Given the fact that there are many models of the Vietnamese delta available to be used, mentioned in the literature (i.e Mike 11 model, TELEMAC, etc) ca you justify why this quasi-2D model was chosen, and what is this model giving different as result as compared with the other available models. The authors are stating

that the research is carried out triggered by the recent discussions in the Vietnamese public and the media. However being a research paper which will not be read by the general public how would such results be conveyed/ made available to the public, or authorities in charge of the system. What is the concern of the public, just that flood pattern is changed?, or are there consequences of such a change?

---

## Referee Comment (RC2) · A. Pathirana (Referee) · 19 May 2017

A. Pathirana (Referee)

assela@pathirana.net

The (worsening) flood hazard situation in the Mekong Delta (as many other Deltaic regions in the world) does not have a single dominant explanation. It is explained different factors like changes in (a) upper boundary flow, (b) lower boundary (sea-level, and tides) and (c) within the delta like urbanization, development of hydraulic structures (e.g. dykes) etc. Depending on the issue at-hand during a given discourse, the dominant explanation usually ends up being one or a few of the above. The perceived increase of flooding at the downstream parts of the Vietnamese Mekong delta, has been explained by different authorities using all of the above explanations. Triggered by large floods in 2011, government entities, media and the public in Vietnam were keen to provide an explanation to seemingly increasing flood hazard in the Delta. In these discussions a dominant reason stated is the building of high-dykes within the

Delta. (Mainly as a means of reducing inundation to pave-way for cultivating 3 crops of rice.)

The authors set-out to investigate this causal link using data analysis techniques and hydro-dynamic modelling. The result is a manuscript that is both interesting and useful not only for the scientist but for the policymakers and other stakeholders of the delta.

First they use flow gauge data to perform statistical analysis of the observed extreme water levels. What they discover is while the temporal trends shown by upstream stations are relatively-small and statistically not-significant, those downstream show stronger trends with high significance. This indicates that the influencing factors for flooding (one or more of (a), (b) and (c) above) has changed. Then they conduct a step-detection analysis which indicates a step-change around year 2000. Taking the inherent uncertainty of the technique (hence the inability to predict exactly where a trends changes), this observation is in-line with the explanation that high-dyke development is a dominant factor for increasing downstream water levels. However, without conducting a series of what-if experiments, it is impossible to say whether high-dyke construction was the ultimate explanation or just a coincidence in time.

They conduct a series of scenario experiments using a quasi-2D flood model (a 1D hydrodynamic model applied in a smart way to represent flooded areas) in order to separate different drivers. As shown in the figure 10., of the manuscript, they demonstrate that 1. High dyke development is a significant contributing factor for increasing floods and high water levels. However, 2. There are other important explanations (Upstream and downstream changes) that has to be taken in to account to explain the full (complex) picture of flood hazard in the Delta.

The analytical work leading to the manuscript is rigorous. The authors employ proper statistical techniques and perform sensitivity and uncertainty analyses where appropriate. The model calibration and validation were done well. However, they should explain the basics of the model they employed in the manuscript. Just referring to the original

source is not adequate here. At least explain the simplifications to the shallow water equation the model employs, solution scheme it uses etc. in the main text (and add an appendix explaining the model in a bit more detail if possible). While the manuscript is generally well written, there are some (minor) language and editorial issues that have to be addressed. Explain what are low-dykes and high-dykes in a way an international reader can readily understand (height limits?). Some colors and patterns used in figure 1 are not clear (at least in the color I wonder whether all the tables included in the main text are really need to be there (why note move the likes of table 5 to an appendix or a supplement?). I do not understand the basis for the last sentence (the recommendation) in the abstract!

---

## Author Comment (AC1) · 29 May 2017

**Response to 1st Referee's Comments on**

**Has dyke development in the Vietnamese Mekong Delta shifted flood hazard downstream?**

**by Nguyen Van Khanh Triet**

The paper presents the interesting case of the flooding in the Vietnamese delta and the consequences of dyke construction on the flooding pattern. The subject well-presented and the approach is detailed. Though as authors are mentioning there should be further studies about the possible causes for the new flooding patterns, the paper presents in depth analysis of the different flooded areas in the downstream of the catchment where large-scale high-dyke development were performed.

1. **I did miss however details of the quasi-2D model used. Can you please give more explanations on the model, not to repeat what is in the reference, but to explain the underlying equations and the solution of them?**

**AUTHORS' REPONSE:** Thank you for the comment. We include the following lines on the model being used and its underlying equation and solution. The 1st paragraph of Sect. 2.4 Hydrodynamic modelling (p.6 – line 18) will be revised as following.

To quantify the impact of the high-dyke development on flood hazard, a hydrodynamic model for the simulation of flood propagation in the MD was used. The model is a quasi-2D model based on the 1D hydrodynamic modelling suite MIKE 11. The MIKE 11 hydrodynamic (HD) module solves the vertically integrated equations of conservation of continuity and momentum (the 'Saint Venant´ equations). The solution of the equations of continuity (1) and momentum (2) is based on an implicit finite difference scheme developed by Abbott and Ionescu (1967). The model domain includes the CFP, the Tonle Sap Lake as well as the majority of the channels and hydraulic structures in the VMD. The model was initially developed by Dung et al. (2011) and refined by Manh et al. (2014). It explicitly takes the complex hydraulic system with intersecting channels and dyked floodplains of the VMD into account. A typical flood compartment, i.e. part of the floodplain encircled by channels and protected by dykes, is described by "virtual" channels with wide cross-sections connected to the channels by sluice gate model structures. These cross-sections were extracted from the available DEM (originally SRTM, now LiDAR DEM). The cross-section width is defined in such a way to preserve the flood compartment area. Dyke-lines of each flood compartment are described by four control structures right after the points where virtual and real channels are linked. These structures are introduced in the model as broad crest weirs. The crest levels of dyke-lines are presented as sill levels of these control structures (see Fig. 2). A comprehensive description of how floodplain compartments are introduced by the "virtual" channels and wide cross-sections can be found in Dung et al. (2011). The model has been calibrated by Dung et al. (2011) and Manh et al. (2014) with recent flood events in the VMD, encompassing the high floods of 2011, the medium floods in 2008 and 2009, and the extraordinary low flood in 2010.

$$\frac{\partial Q}{\partial x} + \frac{\partial A}{\partial t} = q \qquad\qquad (1)$$

$$\frac{\partial Q}{\partial t} + \frac{\partial\left(\alpha\frac{Q^2}{A}\right)}{\partial x} + gA\frac{\partial h}{\partial x} + \frac{gQ|Q|}{C^2 AR} = 0 \qquad (2)$$

where:

| | |
|---|---|
| $Q$ | discharge |
| $A$ | flow area |
| $q$ | lateral inflow |
| $h$ | stage above datum |
| $C$ | Chezy resistance coefficient |
| $R$ | hydraulic or resistance radius |
| $\alpha$ | momentum distribution coefficient |

2. **Given the fact that there are many models of the Vietnamese delta available to be used, mentioned in the literature (i.e. Mike 11 model, TELEMAC, etc.) ca you justify why this quasi-2D model was chosen, and what is this model giving different as result as compared with the other available models.**

**AUTHORS' REPONSE:** Thank you for the comment. We agree there are many hydodynamic models available either open source e.g. TELEMAC, SOBEK or commercial e.g. MIKE. Despite the fact that MIKE model is commercial, the software is widely applied in many scientific studies on the Vietnamese Mekong Delta not limited to hydrodynamics but also to water quality, salinity intrusion and sediment transport. Because the authors' institutes, i.e. GFZ and SIWRR, have purchased licenses for MIKE 11 package, and because the model has been proven to be appropriate for the simulation of the inundation dynamics in the Mekong Delta, this model was chosen for this study. Another benefit of the quasi-2D approach is the fast simulation time which is much faster than for fully 2D models. This enables the simulaition of a number of scenarios within a reasonable time frame.

3. **The authors are stating that the research is carried out triggered by the recent discussions in the Vietnamese public and the media. However being a research paper which will not be read by the general public how would such results be conveyed/ made available to the public, or authorities in charge of the system. What is the concern of the public, just that flood pattern is changed? or are there consequences of such a change?**

**AUTHORS' REPONSE:** Thank you for the comment. This was one of our concerns when selecting a journal for our publication, which should be open-access. Once our paper is accepted for publication in Hydrology and Earth System Science (HESS), freely access is granted to anyone who interested in this issue including scientific communities and decision makers in Vietnam. We do agree that sciencetific publication is hardly read by the general public. Therefore, we plan and hope that we might able to present our findings in a conference/symposium in Vietnam where public press are invited. By doing this our study

might reach a wider reader via pulic media e.g. TV, newspaper. Moreover, the results of the study will be transfered to decision makers, i.e. the relevant authorities in Vietnam through the close contacts between the home institute of the lead author, SIWRR, with these authorities. These contacts are established over a lasrge number of national and international applied and research projects over the last decades. Therefor the "ears of the decision makers" are open to the results and implications of the study.

The main public concerns are (i) how the flood regime downstream of high-dyke areas has changed due to the construction of fully flood protection measures in the northern part of the VMD e.g. at An Giang and Dong Thap. And (ii) should these high-dyke construction to be continued for expansion of tripple rice cropping areas. The findings of our study will provide the nummerical quantification for the first question and can be taken as the basis for decision makers to develop a holistic water and flood risk management plan in the VMD.

**References**

Abbott, M. B., and Ionescu, F.: On the numerical computation of nearly horizontal flows, Journal of Hydraulic Research, 5, 97-117, 1967.

Dung, N. V., Merz, B., Bárdossy, A., Thang, T. D., and Apel, H.: Multi-objective automatic calibration of hydrodynamic models utilizing inundation maps and gauge data, Hydrology and Earth System Sciences, 15, 1339-1354, 2011.

Manh, N. V., Dung, N. V., Hung, N. N., Merz, B., and Apel, H.: Large-scale suspended sediment transport and sediment deposition in the Mekong Delta, Hydrol. Earth Syst. Sci., 18, 3033-3053, 10.5194/hess-18-3033-2014, 2014.

---

## Referee Comment (RC3) · Anonymous Referee #3 · 19 Jun 2017

The paper shows an interesting finding about the correlation between the dyke system and the flooding in Vietnam Mekong Delta by observing the changes in flood characteristics to high-dyke constructions and other possible causes. The paper was well organized and mechanism of the flooding in VMD was explained clearly by the trend analysis and hydrodynamic flood model though there are some other factors than large-scale dykes that can cause the hazard, especially in downstream. The findings of the paper will be very useful not only for the academia but also the local governments in the deltaic regions.

The authors used Monte Carlo experiment for uncertainty analysis of the detected trend. However, it seems that more explanation about the method of analyses measurement error should be clarified, for example, the Reliability Method with median

value and variation of error.

Inundation levels in the Mekong Delta are predominantly determined by ocean tides, sea-level rise, and land subsidence. Although the authors took in account the changes in the tidal dynamics, it is hard to understand well about the tidal propagation model, how it effects to the inundation level in Mekong Delta by tidal harmonic analysis. The land subsidence in VMD is obviously significant following previous research (eg. Erban et al., 2014) and will be the predominant factor leading to more serious floods over the low-lying areas (downstream) (Takagi et al. (2016): Sea-Level Rise and Land Subsidence: Impacts on Flood Projections for the Mekong Delta's Largest City, Sustainability, 8, 959; doi:10.3390/su8090959).

In the conclusion, the authors confirm the claims that the high-dyke development has raised the flood hazard downstream. However, it is not the only and not the most important driver of the observed changes (sea level rise in combination with the widely observed land subsidence and the temporal coincidence of high water levels and spring tides have even larger impacts). Therefore, it should be very carefully considered about the main factor that causes more serious flood in the downstream and need to have more research on this.

---

## Author Comment (AC3) · 26 Jun 2017

**Response to 3rd Referee's Comments on**

**Has dyke development in the Vietnamese Mekong Delta shifted flood hazard downstream?**

**by Nguyen Van Khanh Triet et al.**

We would like to thank the reviewer for taking time and evaluating our work and the manuscript as interesting and useful for the scientific community and the Vietnamese administration. Our answers to the raised comments are as follows:

**General comments:** *The paper shows an interesting finding about the correlation between the dyke system and the flooding in Vietnam Mekong Delta by observing the changes in flood characteristics to high-dyke constructions and other possible causes. The paper was well organized and mechanism of the flooding in VMD was explained clearly by the trend analysis and hydrodynamic flood model though there are some other factors than largescale dykes that can cause the hazard, especially in downstream. The findings of the paper will be very useful not only for the academia but also the local governments in the deltaic regions.*

**AUTHORS' REPONSE:** Thank you for the nice comment

1. **Comment 1.** *The authors used Monte Carlo experiment for uncertainty analysis of the detected trend. However, it seems that more explanation about the method of analyses measurement error should be clarified, for example, the Reliability Method with median value and variation of error.*

**AUTHORS' REPONSE:** Thank you for your comment. However, in this paper our aim is not to analyse the actual measurement errors, which are in fact unknown, but rather show that potential measurement errors do not yield different results. That means we want to show that the results of the trend tests are robust against potential but unknown measurement errors. The range of the potential errors were defined according to typical instrumental errors of water level monitoring instruments plus additional errors caused by human operation and unforeseeable events.

2. **Comment 2.** *Inundation levels in the Mekong Delta are predominantly determined by ocean tides, sea-level rise, and land subsidence. Although the authors took in account the changes in the tidal dynamics, it is hard to understand well about the tidal propagation model, how it effects to the inundation level in Mekong Delta by tidal harmonic analysis.*

**AUTHORS' REPONSE:** Thank you for the comment. The "tidal propagation model" is actually the hydrodynamic model, i.e. the quasi 2D model using MIKE11, which translates the pre-defined lower boundary, i.e. the tidal water levels of the sea, into changing flows and water level in the river system. Because the upper boundary (mekong flow at Kratie) and lower boundary (tidal water levels at the river mouths) are predefined, the hydrodynamic code, which is a numerical representation of the St.-Venant

hydrodynamic equations describing the physics of river flow, calculates the propagation of the tides upstream the river as a consequence of the interplay of tidal water levels and river discharge.

---

## Author Comment (AC2)

**Response to Referee's Comments from Ass. Prof. Assela Pathirana on**

**Has dyke development in the Vietnamese Mekong Delta shifted flood hazard downstream?**

**by Nguyen Van Khanh Triet**

1.  **The analytical work leading to the manuscript is rigorous. The authors employ proper statistical techniques and perform sensitivity and uncertainty analyses where appropriate. The model calibration and validation were done well. However, they should explain the basics of the model they employed in the manuscript. Just referring to the original source is not adequate here. At least explain the simplifications to the shallow water equation the model employs, solution scheme it uses etc. in the main text (and add an appendix explaining the model in a bit more detail if possible).**

**AUTHORS' REPONSE:** Thank you for the possitive comment, which was also raised by reviewer 1. The 1st paragraph of Sect. 2.4 Hydrodynamic modelling (p.6 – line 18) will be revised as following, as also stated in the reply to reviewer 1. For the further detailed information on the MIKE11 packages, the readers might refer to DHI website at the given link https://www.mikepoweredbydhi.com/products/mike-11.

To quantify the impact of the high-dyke development on flood hazard, a hydrodynamic model for the simulation of flood propagation in the MD was used. The model is a quasi-2D model based on the 1D hydrodynamic modelling suite MIKE 11. The MIKE 11 hydrodynamic (HD) module solves the vertically integrated equations of conservation of continuity and momentum (the 'Saint Venant´ equations). The solution of the equations of continuity (1) and momentum (2) is based on an implicit finite difference scheme developed by Abbott and Ionescu (1967). The model domain includes the CFP, the Tonle Sap Lake as well as the majority of the channels and hydraulic structures in the VMD. The model was initially developed by Dung et al. (2011) and refined by Manh et al. (2014). It explicitly takes the complex hydraulic system with intersecting channels and dyked floodplains of the VMD into account. A typical flood compartment, i.e. part of the floodplain encircled by channels and protected by dykes, is described by "virtual" channels with wide cross-sections connected to the channels by sluice gate model structures. These cross-sections were extracted from the available DEM (originally SRTM, now LiDAR DEM). The cross-section width is defined in such a way to preserve the flood compartment area. Dyke-lines of each flood compartment are described by four control structures right after the points where virtual and real channels are linked. These structures are introduced in the model as broad crest weirs. The crest levels of dyke-lines are presented as sill levels of these control structures (see Fig. 2). A comprehensive description of how floodplain compartments are introduced by the "virtual" channels and wide cross-sections can be found in Dung et al. (2011). The model has been calibrated by Dung et al. (2011) and Manh et al. (2014) with recent flood events in the VMD, encompassing the high floods of 2011, the medium floods in 2008 and 2009, and the extraordinary low flood in 2010.

$$\frac{\partial Q}{\partial x} + \frac{\partial A}{\partial t} = q \qquad\qquad (1)$$

$$\frac{\partial Q}{\partial t} + \frac{\partial(\alpha \frac{Q^2}{A})}{\partial x} + gA\frac{\partial h}{\partial x} + \frac{gQ|Q|}{C^2 AR} = 0 \qquad (2)$$

where:

| | |
|---|---|
| $Q$ | discharge |
| $A$ | flow area |
| $q$ | lateral inflow |
| $h$ | stage above datum |
| $C$ | Chezy resistance coefficient |
| $R$ | hydraulic or resistance radius |
| $\alpha$ | momentum distribution coefficient |

2.  **While the manuscript is generally well written, there are some (minor) language and editorial issues that have to be addressed. Explain what are low-dykes and high-dykes in a way an international reader can readily understand (height limits?)**

**AUTHORS' REPONSE:** Thank you for the comment. We have included in our manuscipt a description on height of high-dyke at page 4 and line 4: "crest levels of 4.0-6.0 m.a.s.l." We will additionally include the following text on height limits of low-dyke in the revised manuscipt (page 4 - line 2). "… with a total length of over 13,000 kilometers, of which 8,000 kilometers are low-dyke with crest levels vary from 1.5 – 4.0 m.a.s.l. …"

3.  **Some colors and patterns used in figure 1 are not clear (at least in the color I wonder whether all the tables included in the main text are really need to be there (why note move the likes of table 5 to an appendix or a supplement?).**

**AUTHORS' REPONSE:** Thank you for pointing this out. We suppose it is the pattern which distinguishes the Long Xuyen Quadrangle and the Plain of Reeds that you are mentioning here. We will change the coressponding color and pattern in the revised manuscript (see figure below). Regarding table 5, we do think that it needs to remain in the main text. It presents the basis for our conclusion. Without it, the summarizing results and the conclusions cannot be followed.

[Figure]

**Figure 1.** The Vietnamese Mekong Delta, its flood prone areas and location of measuring stations (red dots). The names in black indicate the provinces in the VMD.

**4. I do not understand the basis for the last sentence (the recommendation) in the abstract!**

**AUTHORS' REPONSE:** This recommendatiojn is base on the finding, that a reduction of 9–13 cm in flood peak might be achieveable if flood water is introduced to those fully protected flood compartments (i.e. areas encircle by high-dyke in the upper part of the delta). Therefor we recommend, analogously to flood management strategies applied in other countries, where Polder areas are deliberately opened for capping of flood peaks and reducing flood hazard downsteam, that flood risk management strategies in the VMD should use the high-dyke areas as retention zones to mitigate the flood hazard downstream.

**References**

Abbott, M. B., and Ionescu, F.: On the numerical computation of nearly horizontal flows, Journal of Hydraulic Research, 5, 97-117, 1967.

Dung, N. V., Merz, B., Bárdossy, A., Thang, T. D., and Apel, H.: Multi-objective automatic calibration of hydrodynamic models utilizing inundation maps and gauge data, Hydrology and Earth System Sciences, 15, 1339-1354, 2011.

Manh, N. V., Dung, N. V., Hung, N. N., Merz, B., and Apel, H.: Large-scale suspended sediment transport and sediment deposition in the Mekong Delta, Hydrol. Earth Syst. Sci., 18, 3033-3053, 10.5194/hess-18-3033-2014, 2014.

---

## Author Response (AR1)

**A. Authors response to the review**

We would like to thank all the reviewers for taking time reading and suggesting ways to improve the manuscript. We found all the comments are very useful. Please find our answers to all raised issues below.

**1. Response to 1st Referee's Comments on "Has dyke development in the Vietnamese Mekong Delta shifted flood hazard downstream?" by Nguyen Van Khanh Triet et al.**

**General comment:** *The paper presents the interesting case of the flooding in the Vietnamese delta and the consequences of dyke construction on the flooding pattern. The subject well-presented and the approach is detailed. Though as authors are mentioning there should be further studies about the possible causes for the new flooding patterns, the paper presents in depth analysis of the different flooded areas in the downstream of the catchment where large-scale high-dyke development were performed.*

**Authors' response:** Thank you for the nice comment.

**Comment 1.1:** *I did miss however details of the quasi-2D model used. Can you please give more explanations on the model, not to repeat what is in the reference, but to explain the underlying equations and the solution of them?*

**Authors' response:** Thank you for the comment. We include the following lines on the model being used and its underlying equation and solution. The 1st paragraph of Sect. 2.4 Hydrodynamic modelling will be revised as following.

To quantify the impact of the high-dyke development on flood hazard, a hydrodynamic model for the simulation of flood propagation in the MD was used. The model is a quasi-2D model based on the 1D hydrodynamic modelling suite MIKE 11. The MIKE 11 hydrodynamic (HD) module solves the vertically integrated equations of conservation of continuity and momentum (the 'Saint Venant´ equations). The solution of the equations of continuity (Eq.1) and momentum (Eq. 2) is based on an implicit finite difference scheme developed by Abbott and Ionescu (1967). The model domain includes the CFP, the Tonle Sap Lake as well as the majority of the channels and hydraulic structures in the VMD. The model was initially developed by Dung et al. (2011) and refined by Manh et al. (2014). It explicitly takes the complex hydraulic system with intersecting channels and dyked floodplains of the VMD into account. A typical flood compartment, i.e. part of the floodplain encircled by channels and protected by dykes, is described by "virtual" channels with wide cross-sections connected to the channels by sluice gate model structures. These cross-sections were extracted from the available DEM (originally SRTM, now LiDAR DEM). The cross-section width is defined in such a way to preserve the flood compartment area. Dyke-lines of each flood compartment are described by four control structures right after the points where virtual and real channels are linked. These structures are introduced in the model as broad crest weirs. The crest levels of dyke-lines are presented as sill levels of these control structures (see Fig. 2). A comprehensive description of how floodplain compartments are introduced by the "virtual" channels and wide cross-sections can be found in Dung et al. (2011). The model has been calibrated by Dung et al. (2011) and Manh et al. (2014) with recent flood events in the VMD, encompassing the high floods of 2011, the medium floods in 2008 and 2009, and the extraordinary low flood in 2010.

$$\frac{\partial Q}{\partial x} + \frac{\partial A}{\partial t} = q \tag{1}$$

$$\frac{\partial Q}{\partial t} + \frac{\partial\left(\alpha\frac{Q^2}{A}\right)}{\partial x} + gA\frac{\partial h}{\partial x} + \frac{gQ|Q|}{C^2 AR} = 0 \tag{2}$$

*where:*

    $Q$                *Discharge*

| $A$ | flow area |
|---|---|
| $q$ | lateral inflow |
| $h$ | stage above datum |
| $C$ | Chezy resistance coefficient |
| $R$ | hydraulic or resistance radius |
| $\alpha$ | momentum distribution coefficient |

**Comment 1.2:** *Given the fact that there are many models of the Vietnamese delta available to be used, mentioned in the literature (i.e. Mike 11 model, TELEMAC, etc.) ca you justify why this quasi-2D model was chosen, and what is this model giving different as result as compared with the other available models.*

**Authors' response:** Thank you for the comment. We agree there are many hydrodynamic models available either open source e.g. TELEMAC, SOBEK or commercial e.g. MIKE. Despite the fact that MIKE model is commercial, the software is widely applied in many scientific studies on the Vietnamese Mekong Delta not limited to hydrodynamics but also to water quality, salinity intrusion and sediment transport. Because the authors' institutes, i.e. GFZ and SIWRR, have purchased licenses for MIKE 11 package, and because the model has been proven to be appropriate for the simulation of the inundation dynamics in the Mekong Delta, this model was chosen for this study. Another benefit of the quasi-2D approach is the fast simulation time which is much faster than for fully 2D models. This enables the simulation of a number of scenarios within a reasonable time frame.

**Comment 1.3:** *The authors are stating that the research is carried out triggered by the recent discussions in the Vietnamese public and the media. However being a research paper which will not be read by the general public how would such results be conveyed/ made available to the public, or authorities in charge of the system. What is the concern of the public, just that flood pattern is changed? or are there consequences of such a change?*

**Authors' response:** Thank you for the comment. This was one of our concerns when selecting a journal for our publication, which should be open-access. Once our paper is accepted for publication in Hydrology and Earth System Science (HESS), freely access is granted to anyone who interested in this issue including scientific communities and decision makers in Vietnam. We do agree that scientific publication is hardly read by the general public. Therefore, we plan and hope that we might able to present our findings in a conference/symposium in Vietnam where public press are invited. By doing this our study might reach a wider reader via public media e.g. TV, newspaper. Moreover, the results of the study will be transferred to decision makers, i.e. the relevant authorities in Vietnam through the close contacts between the home institute of the lead author, SIWRR, with these authorities. These contacts are established over a large number of national and international applied and research projects over the last decades. Therefor the "ears of the decision makers" are open to the results and implications of the study.

The main public concerns are (i) how the flood regime downstream of high-dyke areas has changed due to the construction of fully flood protection measures in the northern part of the VMD e.g. at An Giang and Dong Thap. And (ii) should these high-dyke construction to be continued for expansion of triple rice cropping areas. The findings of our study will provide the numerical quantification for the first question and can be taken as the basis for decision makers to develop a holistic water and flood risk management plan in the VMD.

2. **Response to Referee's Comments from Ass. Prof. Assela Pathirana on "Has dyke development in the Vietnamese Mekong Delta shifted flood hazard downstream?" by Nguyen Van Khanh Triet et al.**

**Comment 2.1:** *The analytical work leading to the manuscript is rigorous. The authors employ proper statistical techniques and perform sensitivity and uncertainty analyses where appropriate. The model calibration and validation were done well. However, they should explain the basics of the model they employed in the manuscript. Just referring to the original source is not adequate here. At least explain the simplifications to the shallow water equation the model employs, solution scheme it uses etc. in the main text (and add an appendix explaining the model in a bit more detail if possible).*

**Authors' response:** Thank you for the positive comment, which was also raised by reviewer 1. The 1st paragraph of Sect. 2.4 Hydrodynamic modelling will be revised as stated in the reply to comment 1.1 of reviewer 1. For the further detailed information on the MIKE11 packages, the readers might refer to DHI website at the given link https://www.mikepoweredbydhi.com/products/mike-11.

**Comment 2.2:** *While the manuscript is generally well written, there are some (minor) language and editorial issues that have to be addressed. Explain what are low-dykes and high-dykes in a way an international reader can readily understand (height limits?)*

**Authors' response:** Thank you for the comment. We have included in our manuscript a description on height of high-dyke at page 3 and line 28: "crest levels of 4.0-6.0 m.a.s.l." We will additionally include the following text on height limits of low-dyke in the revised manuscript (page 3 - line 26 & 27). "… with a total length of over 13,000 kilometers, of which 8,000 kilometers are low-dyke with crest levels vary from 1.5 – 4.0 m.a.s.l. …"

**Comment 2.3:** *While the manuscript is generally well written, there are some (minor) language and editorial issues that have to be addressed. Explain what are low-dykes and high-dykes in a way an international reader can readily understand (height limits?)*

**Authors' response:** Thank you for pointing this out. We suppose it is the pattern which distinguishes the Long Xuyen Quadrangle and the Plain of Reeds that you are mentioning here. We will change the corresponding color and pattern of Figure 1 in the revised manuscript (see figure below). Regarding table 5, we do think that it needs to remain in the main text. It presents the basis for our conclusion. Without it, the summarizing results and the conclusions cannot be followed.

**Comment 2.4:** *I do not understand the basis for the last sentence (the recommendation) in the abstract!*

**Authors' response:** This recommendation is based on the finding, that a reduction of 9–13 cm in flood peak might be achievable if flood water is introduced to those fully protected flood compartments (i.e. areas encircle by high-dyke in the upper part of the delta). Therefor we recommend, analogously to flood management strategies applied in other countries, where Polder areas are deliberately opened for capping of flood peaks and reducing flood hazard downstream, that flood risk management strategies in the VMD should use the high-dyke areas as retention zones to mitigate the flood hazard downstream.

[Figure]

**Figure 1.** The Vietnamese Mekong Delta, its flood prone areas and location of measuring stations (red dots). The names in black indicate the provinces in the VMD.

**3. Response to 3rd Referee's Comments on "Has dyke development in the Vietnamese Mekong Delta shifted flood hazard downstream?" by Nguyen Van Khanh Triet et al.**

**General comments:** *The paper shows an interesting finding about the correlation between the dyke system and the flooding in Vietnam Mekong Delta by observing the changes in flood characteristics to high-dyke constructions and other possible causes. The paper was well organized and mechanism of the flooding in VMD was explained clearly by the trend analysis and hydrodynamic flood model though there are some other factors than largescale dykes that can cause the hazard, especially in downstream. The findings of the paper will be very useful not only for the academia but also the local governments in the deltaic regions.*

**Authors' response:** Thank you for the nice comment.

**Comment 3.1:** *The authors used Monte Carlo experiment for uncertainty analysis of the detected trend. However, it seems that more explanation about the method of analyses measurement error should be clarified, for example, the Reliability Method with median value and variation of error.*

**Authors' response:** Thank you for your comment. However, in this paper our aim is not to analyse the actual measurement errors, which are in fact unknown, but rather show that potential measurement errors do not yield different results. That means we want to show that the results of the trend tests are robust against potential but unknown measurement errors. The

range of the potential errors were defined according to typical instrumental errors of water level monitoring instruments plus additional errors caused by human operation and unforeseeable events.

**Comment 3.2:** *Inundation levels in the Mekong Delta are predominantly determined by ocean tides, sea-level rise, and land subsidence. Although the authors took in account the changes in the tidal dynamics, it is hard to understand well about the tidal propagation model, how it effects to the inundation level in Mekong Delta by tidal harmonic analysis.*

**Authors' response:** Thank you for the comment. The "tidal propagation model" is actually the hydrodynamic model, i.e. the quasi-2D model using MIKE11, which translates the pre-defined lower boundary, i.e. the tidal water levels of the sea, into changing flows and water level in the river system. Because the upper boundary (Mekong flow at Kratie) and lower boundary (tidal water levels at the river mouths) are predefined, the hydrodynamic code, which is a numerical representation of the Saintt.-Venant hydrodynamic equations describing the physics of river flow, calculates the propagation of the tides upstream the river as a consequence of the interplay of tidal water levels and river discharge.

**B. List of all relevant changes made in the manuscript**

Please find the list of changes we made in the revised manuscript according to comments from the reviewers in the below table.

| no. | changes | where changes are made |
|-----|---------|------------------------|
| 1 | explanation of "low-dyke" as recommended by RC2 | Page 3 – Line 25 & 26 |
| 2 | explanation of the basic information of the quasi-2D model being used and the solution scheme as suggestions from RC1 and RC2 | Page 5 – Line 31 to 35 and Page 6 – Line 7 to 16 |
| 3 | adding acknowledgment for comments and suggestions from referee to improve the manuscript. | Page 13 – Line 16 & 17 |
| 4 | changing the colour and pattern in figure 1 for better visualization as recommended by RC2 | Page 20 |

**C. Mark-up manuscript version**

Please find the mark-up version of the manuscript starting from the next page.

[revised manuscript text omitted]